# Age of air as a diagnostic for transport time-scales in global models

Maarten Krol[1,2,3], Marco de Bruine[2], Lars Killaars[4], Huug Ouwersloot[5], Andrea Pozzer[5], Yi Yin[6,7], Frederic Chevallier[6], Philippe Bousquet[6], Prabir Patra[8], Dmitry Belikov[9], Shamil Maksyutov[10], Sandip Dhomse[11], Wuhu Feng[12], and Martyn P. Chipperfield[11,12]

[1]Meteorology and Air Quality, Wageningen University, the Netherlands
[2]Institute for Marine and Atmospheric Research, Utrecht University, the Netherlands
[3]Netherlands Institute for Space Research SRON, Utrecht, the Netherlands
[4]Faculty of Science and Engineering, University of Groningen, the Netherlands
[5]Max-Planck institute for Chemistry, Mainz, Germany
[6]Laboraroire de Sciences du Climat et de l'Environnement (LSCE), Gif sur Yvette, France
[7]now at: Jet Propulsion Laboratory, Pasadena, California, USA
[8]Japan Agency for Marine-Earth Science and Technology (JAMSTEC), Yokohama City, Japan
[9]Hokkaido University, Sapporo, Hokkaido, Japan
[10]Center for Global Environmental Research, National Institute for Environmental Studies, Tsukuba, Ibaraki, Japan
[11]School of Earth and Environment, University of Leeds, United Kingdom
[12]National Centre for Atmospheric Science, University of Leeds, United Kingdom

**Correspondence:** Maarten Krol (maarten.krol@wur.nl)

**Abstract.** This paper presents the first results of an age-of-air (AoA) inter-comparison of six global transport models. Following a protocol, three global circulation models and three chemistry transport models simulated five tracers with boundary conditions that grow linearly in time. This allows for an evaluation of the AoA and transport times associated with inter-hemispheric transport, vertical mixing in the troposphere, transport to and in the stratosphere, and transport of air masses between land and ocean. Since AoA is not a directly measurable quantity in the atmosphere, simulations of $^{222}$Rn and SF$_6$ were also performed. We focus this first analysis on averages over the period 2000–2010, taken from longer simulations covering the period 1988 – 2014. We find that two models, NIES and TOMCAT, show substantially slower vertical mixing in the troposphere compared to other models (LMDZ, TM5, EMAC, and ACTM). However, while the TOMCAT model, as used here, has slow transport between the hemispheres and between the atmosphere over land and ocean, the NIES model shows efficient horizontal mixing and a smaller latitudinal gradient in SF$_6$ compared to the other models and observations. We find consistent differences between models concerning vertical mixing of the troposphere, expressed as AoA differences and modeled $^{222}$Rn gradients between 950 and 500 hPa. All models agree, however, on an interesting asymmetry in inter-hemispheric mixing, with faster transport from the Northern Hemisphere surface to the Southern Hemisphere than vice versa. This is attributed to a rectifier effect caused by a stronger seasonal cycle in boundary layer venting over Northern Hemispheric landmasses, and possibly to a related asymmetric position of the inter-tropical convergence zone. The calculated AoA in the mid-upper stratosphere varies considerably among the models (4 –7 years). Finally, we find that the inter-model differences are generally larger than differences in AoA that result from using the same model with a different resolution or convective parameterisation. Taken together, the AoA model inter-comparison provides a useful addition to traditional approaches to evaluate transport timescales. Results highlight that inter-model differences associated with resolved transport (advection, reanalysis data, nudging) and parameterised transport

(convection, boundary layer mixing) are still large, and require further analysis. For this purpose, all model output and analysis software is available.

## 1 Introduction

The composition of the atmosphere is determined by exchange processes at the Earth's surface, transport processes within the atmosphere, and chemical and physical conversion processes. For example, the atmospheric distribution of the $CH_4$ mixing ratio is driven by natural and anthropogenic emissions at the Earth's surface, atmospheric transport, and removal that is primarily driven by the atmospheric oxidants OH, Cl, and $O(^1D)$ (Myhre et al., 2013). The estimated atmospheric lifetime of $CH_4$ is approximately 9 years (Prather et al., 2012) mainly due to oxidation within the troposphere. Although some $CH_4$ destruction occurs in the stratosphere (Boucher et al., 2009), as witnessed by the decaying mixing ratios with altitude, the slow atmospheric transport from the troposphere to the stratosphere limits its impact on the atmospheric lifetime of $CH_4$. Other atmospheric constituents have widely different budgets. $SF_6$, for instance, has purely anthropogenic sources, mostly in the Northern Hemisphere (NH), and is only broken down in the upper stratosphere (Kovács et al., 2017), resulting in a very long atmospheric lifetime of more than 1000 years. In contrast, $^{222}Rn$ emanates naturally from land surfaces and quickly decays radioactively with a half-life of 3.8 days.

To better understand the changes in the atmosphere, General Circulation Models (GCMs) or Chemistry Transport Models (CTMs) are used to simulate its composition. While CTMs use archived meteorological fields to calculate transport, GCMs calculate their own meteorology. To track the observed state of the atmosphere, the GCMs can be nudged towards meteorological reanalysis data (e.g., Law et al., 2008). Comparing results of these models to atmospheric observations allows assessment of the model performance. Depending on the atmospheric compound, various aspects of atmospheric transport can be investigated. Jacob et al. (1997) used $^{222}Rn$ and other short-lived tracers to evaluate the convective and synoptic-scale transport in CTMs. Since the early 1990s, the atmospheric tracer transport model inter-comparison project (TransCom) has carried out studies to quantify and diagnose the uncertainty in inversion calculations of the global carbon budget that result from errors in simulated atmospheric transport. Initially, TransCom focused on non-reactive tropospheric species such as $SF_6$ (Denning et al., 1999) and $CO_2$ (Law et al., 1996, 2008). A more recent TransCom model inter-comparison (Patra et al., 2011) focused on the ability of models to properly represent atmospheric transport of $CH_4$, focussing on vertical gradients, its average long-term trends, seasonal cycles, inter-annual variations (IAVs) and inter-hemispheric (IH) gradients. The study concluded that models with faster IH exchange for $SF_6$ have smaller IH gradients in $CH_4$. The estimated IH exchange time, calculated based on a time-invariant $SF_6$ emission rate, remained relatively constant over the period of the analysis (1996–2007). Interestingly, a recent study highlighted the importance of IH transport variations in explaining the $CO_2$ mixing ratio difference between Mauna Loa in the NH and Cape Grim, Tasmania, in the Southern Hemisphere (SH) (Francey and Frederiksen, 2016). Specifically, the 0.8 ppm step-like increase of this difference between 2009 and 2010 was attributed to the opening and closing of the upper-tropospheric equatorial westerly duct (Waugh and Funatsu, 2003), with an open-duct pattern from July 2008 to June 2009 (fast IH exchange), followed by closed-duct conditions from July 2009 to June 2010 (slow IH exchange). Confirming this

mechanism, Pandey et al. (2017) found also faster IH transport of $CH_4$ during the strong La Niña in 2011 using the TM5 CTM. The time-scale of IH transport is an important parameter for atmospheric inversion studies. In these studies atmospheric observations are used to infer surface flux magnitude and distribution. For a long-lived greenhouse gas like $CH_4$ fast IH transport implies that a larger fraction of the emissions will be attributed to the NH (Patra et al., 2014, 2011).

The efficiency of models to mix the planetary boundary and to vent emissions to the overlying free atmosphere has been studied using $SF_6$ and has generally revealed too slow mixing over mid-latitude continents in the TM5 (Peters et al., 2004) and MOZART models (Gloor et al., 2007). Together with the convective parameterisation, this determines the rate by which emissions (e.g. $SF_6$ and $^{222}$Rn) are mixed vertically and inter-hemispherically (Locatelli et al., 2015a).

     Another transport timescale relevant for atmospheric composition studies is troposphere-stratosphere exchange (Holton
et al., 1995), driven by the Brewer-Dobson circulation (Butchart, 2014). Depending on the greenhouse gas scenario, global climate model projections predict an acceleration of this global mass circulation of tropospheric air through the stratosphere. Stratospheric AoA and its temporal trend have been determined from $SF_6$ measurements from the MIPAS satellite (Stiller et al., 2012) and from balloon observations (Engel et al., 2009). Using a suite of stratospheric observations, Fu et al. (2015) quantified the acceleration of the Brewer-Dobson circulation as 2.1% per decade for 1980–2009. Other modelling and experimental
studies revealed that the atmospheric composition of the tropopause layer strongly depends on the mixing processes that occur on a wide range of spatiotemporal scales (Berthet et al., 2007; Hoor et al., 2010; Prather et al., 2011; Hsu and Prather, 2014).

     Transport of trace gases in CTMs and GCMs is determined by several factors, for which various choices are possible. First, winds and the choice of advection scheme drive the large-scale dispersion and IH transport of tracers. While CTMs directly use winds from a meteorological reanalysis product, GCMs calculate their own meteorology and optionally apply
a nudging scheme to simulate transport of tracers (e.g., Law et al., 2008). Second, parameterised sub-grid-scale processes such as boundary layer mixing and convection determine vertical gradients and the rate of IH transport (e.g., Locatelli et al., 2015a). Finally, other differences may be caused by the horizontal and vertical grid of the model, and other issues related to spatial and temporal integration. For example, by doubling the vertical resolution of their GCM, Locatelli et al. (2015a) largely improved the representation of $SF_6$ transport from the troposphere to the stratosphere. Similarly, Bândă et al. (2015) reported
large differences in the stratospheric dispersion of the 1991 Pinatubo plume when increasing the vertical resolution in TM5 from 34 to 60 levels.

     To investigate the impact of these choices in the participating models, this paper will present the first results of the TransCom AoA inter-comparison study. The concept AoA originates from stratospheric studies (Hall and Prather, 1993; Hall and Plumb, 1994; Neu and Plumb, 1999; Hall et al., 1999). In brief, the age spectrum in the stratosphere $G(x,t|t_0)$ is calculated as a type
of Green's function that propagates a tropospheric mixing ratio boundary condition into the stratosphere. Given a location $x$ in the stratosphere, $G\delta t$ represents the fraction of air at $x$ that was lost in the troposphere in the time interval between $t - t_0$ and $t - t_0 + \delta t$ (Hall et al., 1999). In practical model applications focussing on stratospheric age spectra, $G(x,t|t_0)$ is calculated as the response of a time-dependent boundary condition $\delta(t - t_0)$ specified in a forcing volume in the troposphere. More recently, this concept has also been applied to tropospheric studies (Holzer and Hall, 2000; Waugh et al., 2013; Holzer
and Waugh, 2015). In this AoA inter-comparison, we will only analyse the mean AoA. In the terminology of Holzer and Hall

**Table 1.** The five AoA tracers and their forcing volume.

| AoA Tracer | Forcing volume |
| --- | --- |
| Surface | Surface < 100 m |
| NHsurface | NH surface <100 m |
| SHsurface | SH surface <100 m |
| Land | Land < 100 m |
| Ocean | Ocean < 100 m |

(2000), AoA is defined as the mean transit time, with the age spectrum being the transit-time probability density function. This property can be easily extracted from model simulations of a passive tracer with linearly growing boundary conditions in a specified atmospheric volume. We will outline the inter-comparison protocol in section 2. The resulting model output focussing on the simulation period 2000–2010 is compared in section 3, and discussed in section 4. Finally, the main conclusions are summarised in section 5.

## 2 Method

### 2.1 AoA protocol

In order to compare transport timescales of a suite of CTMs and GCMs, a protocol was developed that allows a straightforward implementation in existing atmospheric models. We defined five AoA tracers (Table 1) for which linearly growing boundary conditions are applied. These AoA tracers are chosen to study IH transport, transport to the stratosphere, and air mass transport between land and ocean. The mixing ratios of these tracers in the models are initialised to zero on January 1, 1988 and simulations are run to the end of 2014.

According to the protocol, the mixing ratio of each AoA tracer in its respective forcing volume is set every time step to a value $B = f \times t$, with $t$ the elapsed time (in s) since January 1988, and $f$ a forcing constant of $1 \times 10^{-15}$ mol mol$^{-1}$s$^{-1}$. As a result, the mixing ratio in the forcing volume will have a value of 852.0768 nmol mol$^{-1}$ at the end of the simulation, and lower values elsewhere. In theory, numerical issues with advection might lead to small wiggles in the vicinity of strong mixing ratio gradients. This may result in small unphysical negative mixing ratios that cannot be handled by some model transport schemes. To remedy this, models may be initialised with a uniform 100 nmol mol$^{-1}$ initial condition, a value that is subtracted before further analysis.

Modellers are requested to calculate the exact fraction of each grid box within the forcing volume. This calculation involves the land mask, the fraction of the grid box in either the NH and SH, and the geopotential height. The protocol provides

example code to help with the implementation. Importantly, the mixing ratio of the grid boxes within the forcing volume was set according to:

$$X_{new} = f_{set} \times B + (1 - f_{set}) \times X_{old}, \tag{1}$$

where $f_{set}$ is the fraction of the grid box within the forcing volume, and $X_{old}$ the mixing ratio before the forcing procedure.
Note that in locations where $f_{set} = 0$, nothing needs to be changed, and the mixing ratio changes are purely driven by transport from the forcing volume. To diagnose AoA, the simulated mixing ratios $X$ in the atmosphere can be converted into AoA by $L = t - \frac{X}{f}$, with $L$ the AoA in seconds and $t$ the elapsed time of the simulation.

Figure 1 shows the simulated mixing ratios and applied forcing for the tracer "Surface" at the location of the Cabauw tower from May 6 to May 15, 2010, together with the derived AoA. We use here output from the TM5 model on $1 \times 1$ degree resolution (see Table 4). At 20 m above the surface during night-time AoA is generally close to zero, due to the fact that the volume is forced in the lowest 100 m, and vertical mixing is limited in a stable nocturnal boundary layer. During daytime, however, older air from aloft is mixed in, and the AoA increases depending on the depth of the mixing layer and the strength of the vertical mixing. At 200 m, outside the forcing volume, air is generally older. During some nights, e.g. from May 8 to May 9, the surface gets decoupled from the air masses aloft, signalling a stable boundary layer. Note that the depth of the lowest model layer is approximately 20 m, which implies that this layer is entirely within the forcing volume (the applied forcing is indicated by the blue line). However, TM5 still calculates a non-zero AoA in the lowest model layer, because the mixing ratios are sampled after vertical transport, which mixes in older air (Krol et al., 2005). We do not provide recommendations on the sampling strategy. For instance, models that sample right after forcing, simulate a linear mixing ratio increase at 20 m (blue line in the upper panel of Figure 1) and a constant and zero AoA at 20 m (blue line in the lower panel of Figure 1). The current paper will focus on large-scale transport timescales and we found no major influence of this sampling strategy on the results.

To complement the AoA tracers, modellers were also requested to simulate the tracers listed in Table 2. Simulations of $^{222}$Rn are intended to study vertical and synoptic-scale transport (Jacob et al., 1997); SF$_6$ is used to diagnose IH transport, stratosphere–troposphere exchange, and stratospheric AoA; E90 is used to diagnose the "chemical" tropopause and will be compared to Prather et al. (2011). Note that we also included a $^{222}$Rn simulation with monthly varying emissions over Europe during 2006–2010, based on the high resolution emission maps presented in Karstens et al. (2015). The current paper will, however, not analyse these simulations.

Homogeneous surface emissions (in kg m$^{-2}$ s$^{-1}$) of E90 (E$_{E90}$) are calculated such that the mean steady-state atmospheric mixing ratio of E90 will approach 100 nmol mol$^{-1}$:

$$E_{E90} = \frac{M_{atm} \times 100 \times 10^{-9}}{\tau_{E90} \times 4\pi r^2} \tag{2}$$

with $M_{atm}$ the atmospheric mass (kg), $r$ the radius of the Earth (m), and $\tau_{E90}$ the e-folding lifetime (90 days in units s) of E90.

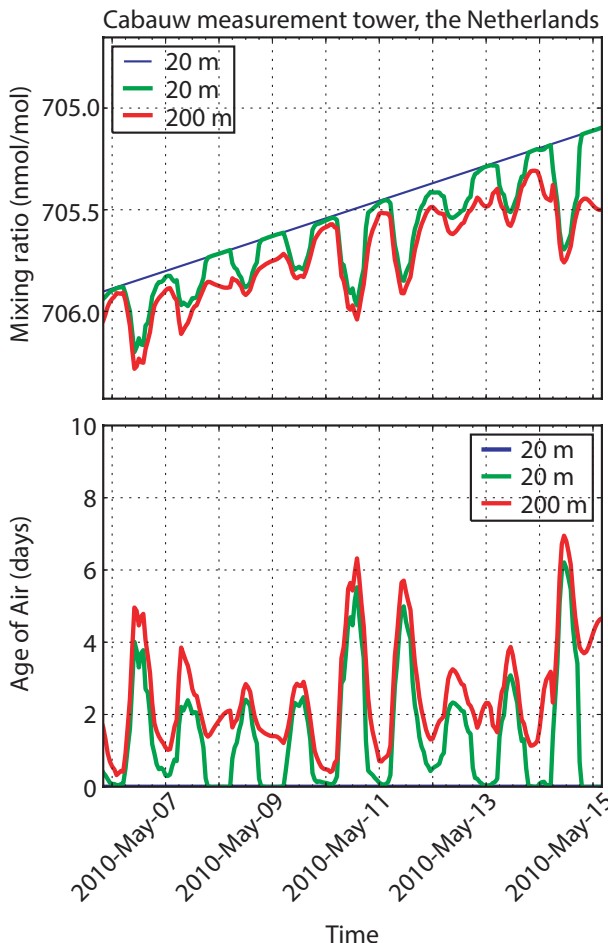

**Figure 1.** Upper panel: Mixing ratios of an AoA tracer simulated by the TM5 model (TM5_1x1) forced at the surface along the Cabauw measurement tower in the Netherlands during May 2010. The blue line represents the applied forcing. Lower panel: Similar, but transferred into AoA.

Emissions of $SF_6$ based on EDGAR4.0 with corrections suggested by Levin et al. (2010) are given in Table 3. The emission distribution is similar to Patra et al. (2011).

NetCDF-formatted input files were provided with emissions on $1 \times 1$ degree resolution and initial $SF_6$ mixing ratios for January 1, 1988. In terms of output, modellers were requested to provide monthly mean 3D mixing ratios and hourly mixing ratio time series at 247 atmospheric measurement stations (interpolated or grid-value). Furthermore, hourly atmospheric profiles of mixing ratios and meteorological variables (u-wind, v-wind, surface pressure, boundary layer height, geopotential height) were requested at 119 locations. In this first analysis we will concentrate on the output of monthly mean mixing ratios that have been provided by three CTMs and three GCMs, some of them running in different configurations. Table 4 lists the participating models along with their horizontal and vertical resolution. Three models (LMDZ, EMAC, ACTM) run in "on-line"

**Table 2.** Additional tracers in the model inter-comparison.

| Tracer | Remarks |
|---|---|
| $^{222}Rn$ | Radon tracer, similar to (Jacob et al., 1997; Law et al., 2008; Patra et al., 2011) |
| $^{222}RnE$ | As $^{222}Rn$, but with specific monthly emissions over Europe in 2006-2010 (Karstens et al., 2015) |
| $SF_6$ | $SF_6$ tracer, similar to Patra et al. (2011), but with updated yearly emissions (Table 3) |
| E90 | Tracer with surface emissions and atmospheric lifetime of 90 days (Prather et al., 2011) |

**Table 3.** Yearly emissions of $SF_6$ for the period 1988-2015.

| Year | Source (mmol s$^{-1}$) | Year | Source (mmol s$^{-1}$) |
|---|---|---|---|
| 1988 | 934 | 2002 | 1223 |
| 1989 | 938 | 2003 | 1258 |
| 1990 | 1036 | 2004 | 1268 |
| 1991 | 1116 | 2005 | 1299 |
| 1992 | 1210 | 2006 | 1366 |
| 1993 | 1303 | 2007 | 1475 |
| 1994 | 1381 | 2008 | 1555 |
| 1995 | 1392 | 2009 | 1577 |
| 1996 | 1312 | 2010 | 1599 |
| 1997 | 1208 | 2011 | 1642 |
| 1998 | 1162 | 2012 | 1685 |
| 1999 | 1177 | 2013 | 1729 |
| 2000 | 1201 | 2014 | 1772 |
| 2001 | 1197 | 2015 | 1816 |

mode, meaning that meteorology is calculated by the physics module of the model, and nudged towards a reanalysis dataset, such as ERA-Interim (Dee et al., 2011), provided by European Centre for Medium-Range Weather Forecasts and JRA-25 from the Japanese Meteorological Agency (Onogi et al., 2007; Chiaki and Toshiki, 2016). Three other models (TM5, NIES, TOMCAT) run in "off-line" mode, and directly read in meteorological driver data from a reanalysis. TM5_EC-Earth reads in

5    meteorological data from the EC-Earth model (Hazeleger et al., 2010) that is nudged to ERA-Interim (see Section 2.3)

In the next subsections, specific information on the participating models and the measurements is given.

**Table 4.** Short-hand notation of the models participating in this study, along with model information.

| Model submission | Base model | longitude × latitude | Vertical levels | Meteorological Driver Data |
|---|---|---|---|---|
| LMDZ3 | LMDZ | $3.75° \times 1.875°$ | 39 hybrid $\sigma$-pressure | Nudged to ERA-Interim |
| LMDZ5A | LMDZ | $3.75° \times 1.875°$ | 39 hybrid $\sigma$-pressure | Nudged to ERA-Interim |
| TM5_3x2 | TM5 | $3° \times 2°$ | 60 hybrid $\sigma$-pressure | ERA-Interim |
| TM5_1x1 | TM5 | $1° \times 1°$ | 60 hybrid $\sigma$-pressure | ERA-Interim |
| TM5_EC-Earth | TM5 | $3° \times 2°$ | 34 hybrid $\sigma$-pressure | EC-Earth, nudged to ERA-Interim |
| EMAC_T63 | EMAC | $\approx 1.875° \times 1.875°$ | 90 hybrid $\sigma$-pressure | Nudged to ERA-Interim |
| EMAC_T106 | EMAC | $\approx 1.125° \times 1.125°$ | 90 hybrid $\sigma$-pressure | Nudged to ERA-Interim |
| ACTM | ACTM | $\approx 2.81° \times 2.81°$ | 67* $\sigma$ up to 90 km | Nudged to JRA-25 |
| NIES | NIES | $2.5° \times 2.5°$ | 32 hybrid $\sigma - \theta$ up to 5 hPa | JRA-25 |
| TOMCAT | TOMCAT | $\approx 2.81° \times 2.81°$ | 60 hybrid $\sigma$-pressure | ERA-Interim |

*Only the lower 50 layers have been submitted

## 2.2 LMDZ

LMDz, developed at Laboratoire de Météorologie Dynamique with "z" standing for zoom capacity, is the global general circulation model of the IPSL Earth system model (Hourdin et al., 2006, 2012). Here, we include two offline versions of LMDz – LMDZ3 and LMDZ5A –, both with a horizontal resolution of $1.875°$ (latitude) $\times 3.75°$ (longitude) and a vertical resolution of 39 hybrid sigma-pressure levels, commonly chosen for global inverse studies using LMDz (Chevallier, 2015; Locatelli et al., 2015b; Yin et al., 2017). These two versions are different in terms of physical parameterisation of deep convection and boundary layer mixing. LMDZ3 uses the deep convection scheme of Tiedtke (1989) and the boundary layer mixing scheme of Louis (1979). LMDZ5A is an updated version that uses the deep convection scheme of Emanuel (1991) and the boundary layer mixing parameterisation from Louis (1979), further adjusted according to Deardorff (1966). More details regarding the configurations of the physics and comparison to other versions are described in Locatelli et al. (2015a). Sea surface temperature and sea ice coverage from the ERA-Interim are used as boundary conditions. Horizontal winds are nudged towards the ERA-Interim wind fields with a relaxation time of 3 hours.

## 2.3 TM5

The TM5 model (Krol et al., 2005) has different application areas with a common core. TM5 allows for a flexible grid definition with two-way nested zoom regions. This version is mainly used in inverse modelling applications, focussing on $CO_2$ (Peters et al., 2010), $CH_4$ (Bergamaschi et al., 2013; Houweling et al., 2014), and CO (Krol et al., 2013). The chemistry version of TM5 (Huijnen et al., 2010) was recently adapted for massive parallel computing (Williams et al., 2017). The AoA simulations were conducted in this so-called TM5-MP version.

The TM5 model is used in three versions that differ in horizontal resolution and in the way meteorological data is used. The version named TM5_3x2 simulated the AoA experiment at a resolution of $3° \times 2°$ (longitude × latitude). In the version TM5_1x1, a horizontal resolution of $1° \times 1°$ was used. TM5_1x1 and TM5_3x2 refer to simulations with the off-line TM5-MP version that reads in the meteorological fields from files. Like earlier model versions, TM5_1x1 and TM5_3x2 are driven by the ERA-Interim and updated every 3 h, with time-interpolation during time integration. For this inter-comparison, we include all 60 vertical levels provided by the ECMWF ERA-Interim reanalysis. Convective mass fluxes are derived from entrainment and detrainment rates from the ERA-Interim dataset. This replaces the convective parameterisation used in the previous TransCom inter-comparison (Patra et al., 2011), which was based on Tiedtke (1989). According to Tsuruta et al. (2016) the mass fluxes produced with the ERA-Interim data set lead to faster inter-hemispheric transport compared to the old model version using the Tiedtke (1989) scheme that was used in the earlier TransCom study (Patra et al., 2011).

The vertical diffusion in the free troposphere is calculated according to Louis (1979), and in the boundary layer by the approach of Holtslag and Boville (1993). Diurnal variability in the boundary layer height is determined using the parameterisation of Vogelezang and Holtslag (1996). Advective fluxes are calculated using the slopes scheme (Russel and Lerner, 1981), with a refinement in time step whenever the Courant–Friedrichs–Lewy (CFL) criterion is violated.

TM5_EC-Earth is the atmospheric transport model of the Earth System Model EC-Earth version 3.2.2 (Version 2.3 is described in Hazeleger et al. (2010, 2012)) The main differences between the two versions are an increased resolution (from T159 with 62 vertical layers in version 2.3 to T255 with 91 vertical layers in version 3.2.2) and updated versions of the separate model components. Only the atmospheric components are used in this study. The dynamical core of EC-Earth is based on the Integrated Forecasting System (IFS), version cy36r4 (Molteni et al., 2011). In contrast to the offline versions of TM5, the dynamical properties of the atmosphere are simulated by EC-Earth in TM5_EC-Earth. EC-Earth was nudged towards ERA-Interim data for temperature, vorticity, divergence, and the logarithm of the surface pressure. The nudging used a relaxation time of 6 hours, allowing for continuous atmospheric dynamics when following the ERA-Interim data. Every 6 hours, the meteorological data is transferred via de OASIS3 coupler (Valcke, 2013) to TM5-MP, which uses this data to calculate the actual atmospheric transport as described above. Apart from the meteorological data, the model code of TM5-ECEarth is similar to TM5_3x2, but with a reduced number of vertical levels (34, i.e. a subset of the 91 levels used in IFS, instead of 60).

## 2.4 EMAC

The EMAC (ECHAM/MESSy Atmospheric Chemistry) model employed in this study, combines an updated version 2.50 of the MESSy (Modular Earth Submodel System) framework (Jöckel et al., 2005, 2010) with version 5.3.02 of the ECHAM5 (European Centre Hamburg) general circulation model (Roeckner et al., 2006). The version of the EMAC model used here, first described by Jöckel et al. (2006), incorporates a recent update to convective transport of tracers (Ouwersloot et al., 2015) and is further improved upon to facilitate nudging of AoA tracers. These modifications are included in version 2.52 of MESSy.

Dynamical properties are simulated by EMAC itself. The model dynamics are weakly nudged in the spectral space, nudging temperature, vorticity, divergence and surface pressure (Jeuken et al., 1996). Different nudging coefficients are used for different vertical levels, with no nudging in the boundary layer and above ~10 hPa, and with maximum nudging in the free

troposphere. Convective mass fluxes are diagnosed in the CONVECT submodel (Tost et al., 2006), while the resulting transport is calculated by the CVTRANS model (Tost et al., 2010; Ouwersloot et al., 2015). The simulations with EMAC make use of 90 vertical hybrid sigma pressure levels. Data is available for two horizontal resolutions: T63 ($192 \times 96$ grid) with a fixed time step of 6 min, and T106 ($320 \times 160$ grid) with a fixed time step of 4 min.

## 2.5 ACTM

The CCSR/NIES/FRCGC (Center for Climate System Research/National Institute for Environmental Studies/Frontier Research Center for Global Change) atmospheric general circulation model (AGCM)-based chemistry-transport model (ACTM) has been developed for simulations of long-lived gases in the atmosphere (Numaguti et al., 1997; Patra et al., 2009a, 2014). The ACTM simulations are performed at a horizontal resolution of T42 spectral truncation (~2.8 × 2.8 deg) with 67 sigma levels in the vertical and model top at ~90 km. The horizontal winds and temperature of ACTM are nudged with JRA-25. The nudging forces the AGCM-derived meteorology towards the reanalysed horizontal winds ($u$ and $v$ components) and temperature (T) with relaxation times of 2 and 5 days, respectively, except for the top and bottom model layers.

The heat and moisture exchange fluxes at the Earth's surface are calculated using inter-annually varying monthly-mean sea ice and sea-surface temperature from the Hadley Centre observational data products (Rayner et al., 2003). Tracer advection is performed using a 4th order flux-form advection scheme comprising of monotonic piecewise parabolic method (Colella and Woodward, 1984) and a flux-form semi-Lagrangian scheme (Lin and Rood, 1996).

Sub-grid-scale vertical fluxes are approximated using a non-local closure scheme based on Holtslag and Boville (1993) and the level 2 scheme of Mellor and Yamada (1974). The cumulus parameterisation scheme is simplified from Arakawa and Schubert (1974) and used for calculating the updraft and downdraft of tracers by cumulus convection.

## 2.6 NIES

The National Institute for Environmental Studies (NIES) Eulerian three-dimensional offline transport model is driven by the JRA-25 dataset. It employs a reduced horizontal latitude–longitude grid with a spatial resolution of $2.5° \times 2.5°$ near the equator (Maksyutov and Inoue, 1999) and a flexible hybrid sigma–isentropic ($\sigma$–$\theta$) vertical coordinate, which includes 32 levels from the surface up to 5 hPa (Belikov et al., 2013b). The vertical transport for all levels above tropopause and higher than a potential temperature level of 295 K follows a climatological adiabatic heating rate. The parametrisation of turbulent diffusivity separates transport processes in the planetary boundary layer (provided by the ECMWF ERA-Interim reanalysis) from the free troposphere following the approach by Hack et al. (1993). Vertical mass fluxes due to cumulus convection are based on the convective precipitation rate provided by the reanalysis dataset (Austin and Houze Jr, 1973; Belikov et al., 2013a). A modified Kuo-type parameterization scheme (Grell et al., 1994) is used to set cloud top and cloud bottom height. Transport by convective updrafts and downdrafts includes entrainment and detrainment processes as described by Tiedtke (1989).

## 2.7 TOMCAT

TOMCAT/SLIMCAT is a global 3-D off-line chemical transport model (Chipperfield, 2006). It is used to study a range of chemistry-aerosol-transport issues in the troposphere (e.g., Monks et al., 2017) and stratosphere (e.g., Chipperfield et al., 2015). The model is usually forced by ERA-interim, although GCM output can also be used. When using ECMWF fields, as in the experiments described here, the model reads in the 6-hourly fields of temperature, humidity, vorticity, divergence and surface pressure. The resolved vertical motion is calculated online from the vorticity. The model has different options for the parameterisations of sub-grid-scale tracer transport by convection (Stockwell and Chipperfield, 1999; Feng et al., 2011) and boundary layer mixing (Louis, 1979; Holtslag and Boville, 1993). Tracer advection is performed using the conservation of second-order moments scheme of Prather et al. (1987). For the experiments the model was run at horizontal resolution of $2.8°$ $\times$ $2.8°$ with 60 hybrid $\sigma$-pressure levels from the surface to ~60 km. These follow the vertical levels from the meteorological fields from ERA-Interim, which are used to force the model. Convective mass fluxes were diagnosed online using a version of the Tiedtke scheme (Stockwell and Chipperfield, 1999) and mixing in the boundary layer is based on the local scheme of Louis (1979). Previous work with the model has shown that these schemes tend to underestimate the mixing out of the boundary layer and convective transport to the upper troposphere (e.g. Feng et al., 2011) but these were the options available for the multi-decadal runs analysed here.

## 2.8 Measurement data

We will compare model results to latitudinal $SF_6$ gradients measured by the National Oceanic & Atmospheric Administration Earth System Research Laboratory (NOAA/ESRL) (Hall et al., 2011). We use the combined data set constructed from flask data measured by the Halocarbons and other Atmospheric Trace Species (HATS) group and hourly Chromatograph for Atmospheric Trace Species (CATS) data (downloaded from https://www.esrl.noaa.gov/gmd/hats/combined/SF6.html), and include only stations with a full measurement record in the period 2000–2011. To account for model-data offsets, 11 year time series with monthly resolution were constructed of the stations' mixing ratios with respect to the South Pole. The mean and standard deviations of these time series are added to the modeled latitudinal gradients below.

In Appendix A we further compare the simulated $SF_6$ latitudinal and vertical gradients to measurements made during the High-performance Instrumented Airborne Platform for Environmental Research (HIAPER) Pole-to-Pole Observation (HIPPO) campaigns (Wofsy et al. , 2011), similar to comparisons presented in Patra et al. (2014).

## 3   Results

### 3.1   Tropospheric AoA

First, we focus on zonal and multi-year averages to investigate differences among the models in IH and vertical transport in the troposphere. We created these averages by (i) averaging the monthly mean mixing ratios of the participating models zonally

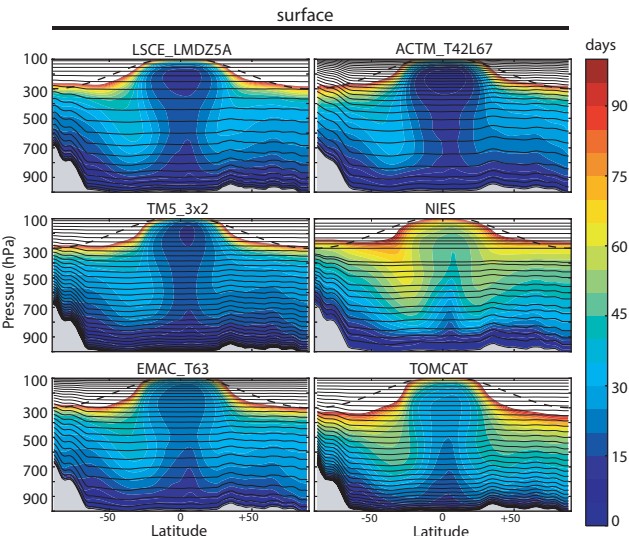

**Figure 2.** Zonally averaged AoA (days) in the troposphere for tracer "Surface". The results have been averaged over the period 2000–2010 (11 years). The light grey areas correspond to the zonal mean orography in the models. The thin black lines denote the mid-pressure levels of the models. For reference, the dotted black line in all panels denotes the climatological tropopause (Lawrence et al., 2001). The white areas correspond to areas in which the air is older than 100 days.

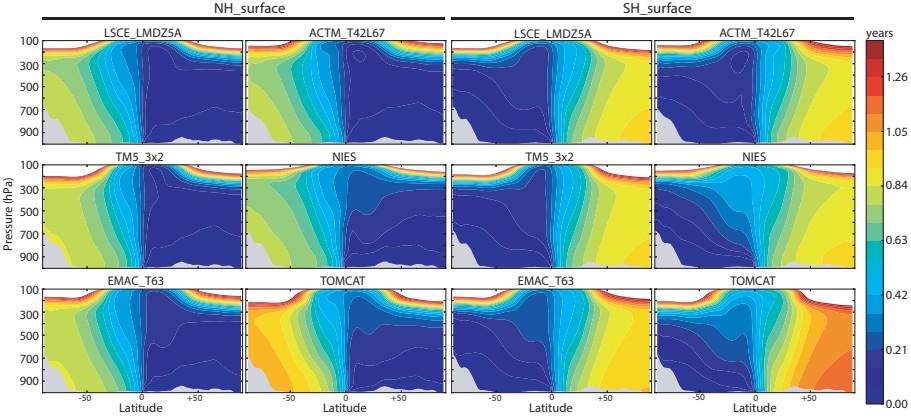

**Figure 3.** Zonally averaged AoA (years) in the troposphere for tracers "NHsurface" (left) and "SHsurface" (right). The results have been averaged over the period 2000-2010 (11 years). The light grey areas correspond to the zonal mean orography in the models. The white areas correspond to regions in which the air is older than 1.4 years.

and over time (ii) converting the mean mixing ratios to AoA. These latitude–pressure cross-sections are presented in Figure 2 for AoA tracer Surface, and in Figure 3 for AoA tracers NHsurface and SHsurface (see Table 1).

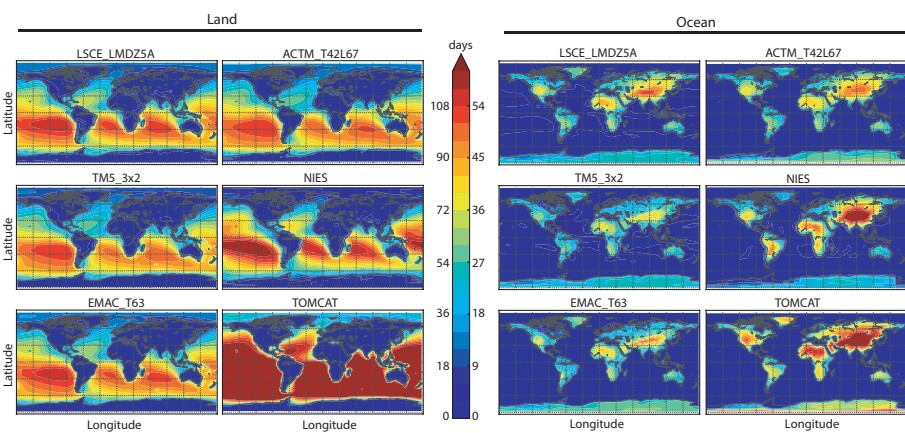

**Figure 4.** Latitude-longitude plot of the AoA (days) derived from tracers "Land" (left) and "Ocean" (right), evaluated in the lowest model layer. The results have been averaged over the period 2000-2010 (11 years). Ages older than 120 days (land tracer) or 60 days (ocean tracer) appear in dark red.

The AoA in the troposphere derived from tracer "Surface" is generally younger than 100 days (Figure 2). The 100 day contour sharply marks the transition to the stratosphere at around 300 hPa at the poles. In the tropics this transition is located at a pressure lower than 100 hPa, which generally agrees with the definition of the climatological tropopause given by Lawrence et al. (2001). Deep convective transport drives the vertical transport in the tropics, and this deep convective region is bounded

by tongues of older air, signalling intrusions of stratospheric air (Holton et al., 1995). Although all models agree on this general pattern, clear differences are also present. Deep convective mixing in the tropics is strongest in ACTM, while TOMCAT and NIES generally show slower vertical trnasport and larger AoA gradients between the surface and the upper troposphere.

Tracers "NHsurface" (Figure 3, left) and "SHsurface" (Figure 3, right) are used to diagnose IH transport. Generally, the air at high latitudes has an age between 0.6 and 1.2 years in the hemisphere opposite to the forcing volume. Here, all models

agree on an interesting asymmetry: AoA derived from "SHsurface" around the North Pole is 0.10 – 0.17 years older than AoA derived from "NHsurface" around the South Pole. TOMCAT, diagnosed with slow vertical mixing, has older air around at high latitudes, while NIES (also slow vertical mixing) has high-latitude ages more in line with other models, indicating fast horizontal transport near the surface. Figure 3 illustrates the fact that exchange of air between the hemispheres proceeds faster at higher altitudes (200-500 hPa) (Prather et al., 1987), and consequently steep AoA gradients are observed close to the surface

in the tropics.

## 3.2 AoA derived from "Land" and "Ocean" tracers

Figure 4 shows AoA derived from tracers "Land" (left) and "Ocean" (right) evaluated in the lowest model level. Oldest air related to "last contact with land" is found over the Southern Oceans, with ages older than 100 days. Models generally agree on the fact that old air is found in the stationary high-pressure areas over the ocean that are related to the Hadley and Walker

circulations. This agrees qualitatively with previous findings using the E90 tracer (Prather et al., 2011). The TOMCAT model,

in the configuration used here, has systematically larger AoA at the surface. Dominant land-ocean transport patterns are easily discerned from the simulations with the "Land" tracer. Young air over oceans is found south-east of South America, north-west of Australia, and towards the north-east of North-America and Asia, associated with the position of the upper-air jet-stream. The oldest air in the Southern East Pacific for tracer "Land" indicates that this region is most isolated from the NH emissions. In accordance with this, the lowest $CH_4$ mixing ratios on the Earth's surface are found at the NOAA site Easter Island (EIC). Patra et al. (2009b) attributed these to "old" air in combination with strong removal of $CH_4$ by OH at tropical latitudes.

The AoA derived from tracer "Ocean" (right-hand panels in Figure 4) generally shows ages less than 60 days over land, with the older ages logically located in deep inland areas. Its AoA over land is strongly determined by dominant circulation patterns such as the monsoon, trade winds, and jet-streams. Although these patterns are similar in all models, the spread is considerable, with TOMCAT again showing the oldest air, and TM5 showing the youngest air over land surfaces. These differences are related to the boundary layer parameterisations in the models, and in the case of TOMCAT the use of the simple local Louis (1979) scheme. Earlier studies (Wang et al., 1999; Chipperfield, 2006) revealed that this type of local boundary layer (BL) mixing scheme leads to slower exchange between the BL and the free atmosphere.

### 3.3 Stratospheric AoA

We use tracer "Surface" to compare the simulated stratospheric AoA. Here it should be noted that differences in tropospheric mixing influence the results. For instance, the TOMCAT and NIES AoA are systematically older at the tropopause (see Figure 2). Figure 5 shows the stratospheric AoA for all models from 100 hPa to the top of the atmosphere, averaged over the period 2000–2010 (11 years). As expected, oldest AoA is found at the high-altitude poles. Considerable model spread is found with the oldest air (up to 7 years) in LMDZ5A and the youngest air in ACTM (< 5 years). The transport in EMAC, TOMCAT, LMDZ, and TM5 is driven by – or nudged to – ERA-Interim meteorology (see Table 4), and one would expect similar stratospheric AoA in these models. However, AoA in the stratosphere is not only determined by the driving meteorological data, but also by the treatment of advection (specifically vertical transport), nudging parameters, and the number of vertical layers in the model (Prather et al., 2008). According to a stratospheric AoA study by Diallo et al. (2012) the use of instantaneous wind fields at 3 or 6-hourly time intervals in CTMs may under-sample fast vertical variations during these time intervals. For GCMs that are nudged to reanalysis meteorological data, gravity wave noise may influence the stratospheric AoA. Finally, the vertical coordinate system may influence numerical transport effects (Chipperfield, 2006; Diallo et al., 2012). Other studies into stratospheric AoA (Garny et al., 2014; Ploeger et al., 2015) highlighted the importance to separate stratospheric mixing between a (slow) residual circulation and (fast) eddy mixing. Further analysis on the stratospheric AoA in this model ensemble is, however, left for future exploration.

### 3.4 Interhemispheric transport

In this section, we compare the inter-hemispheric transport times using tracers "NHsurface" and "SHsurface" to the simulated latitudinal gradients of $SF_6$. To this end, we show in Figure 6 ( panel (a) ) the simulated zonal average latitudinal $SF_6$ gradient at the surface. All gradients have been scaled relative to the South Pole $SF_6$ mixing ratio. Also included in the figure are

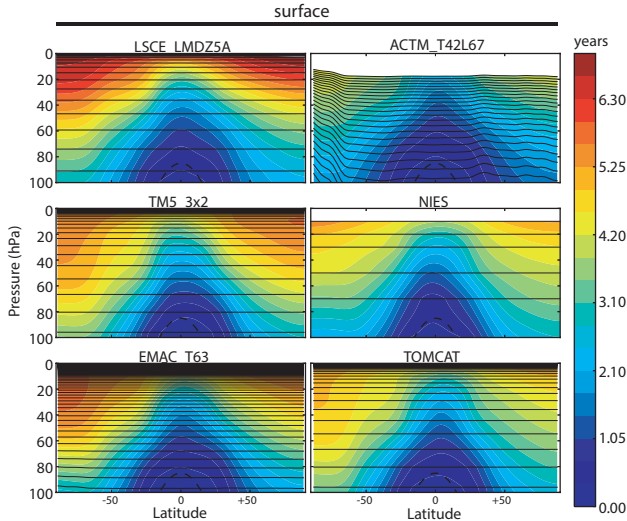

**Figure 5.** Zonally averaged AoA (years) in the stratosphere for tracer "Surface". The results have been averaged over the period 2000-2010 (11 years). The thin black lines denote the mid-pressure levels of the model. For the ACTM model, this level is at around 17 hPa for the uppermost $50^{th}$ layer. The dotted black line denotes the climatological tropopause. Note the linear y-scale that starts at 100 hPa.

the latitudinal gradients measured by NOAA/ESRL (Hall et al., 2011). Since the selected model results are averages over all longitudes, including the land masses with high emissions, modeled zonal averages exceed the observations at NH mid-latitudes. Most notably, high altitude stations such as Niwot Ridge (3523 m above sea level (masl)) and Mauna Loa (3397 masl) show much smaller mixing ratio differences with respect to the South Pole than, for example, Cape Kumukahi at sea level.

This is confirmed by panel (b) of Figure 6, in which the modeled $SF_6$ fields are zonally averaged over the Pacific Ocean only (from 150°E to 210°E). In this average all models except NIES agree well within 0.1 pmol mol$^{-1}$ on the latitudinal gradient. Due to too reduced vertical mixing combined with fast latitudinal transport, the NIES model underestimates the latitudinal $SF_6$ gradient. The TOMCAT model simulates the observed latitudinal gradient of $SF_6$ on clean background stations well, but also shows high accumulation over land (see Figures 4 and 6a). In Appendix A we further compare the simulated $SF_6$ latitudinal

and vertical gradients to measurements made during the HIPPO campaigns (Wofsy et al. , 2011).

Panel (c) of Figure 6 depicts a composite of the AoA of tracer "NHsurface" at the SH surface, and of tracer "SHsurface" at the NH surface. Again, all results are zonal averages over 2000–2010 (11 years) and include all longitudes. Panel (d) of Figure 6 confirms the NH–SH asymmetry noted earlier in Section 3.1, but also makes it clear that the asymmetry differs with model. Panel (d) of Figure 6 highlights the NH–SH asymmetry by subtracting the "NHsurface" AoA sampled at the SH surface from

15 the "SHsurface" AoA sampled at the NH surface. In all models, differences grow gradually from 10° latitude to the pole, where the AoA differences range between 0.10 year (TM5_3x2) and 0.17 year (EMAC_T106). The model versions with higher spatial resolution (TM5_1x1, EMAC_T106) show larger AoA differences compared to the lower resolution versions. The TM5_EC-Earth version differs from the other TM5 versions, likely because the physics of the EC-Earth model (IFS version cy36r4) leads

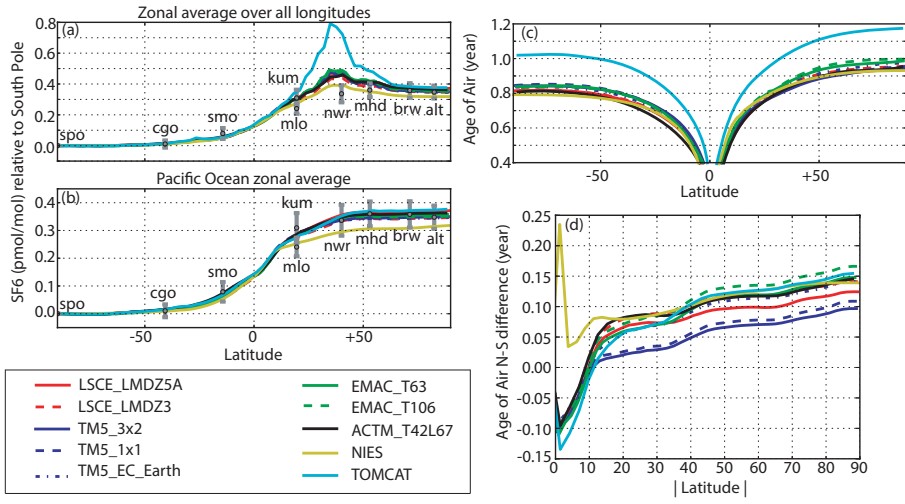

**Figure 6.** Panels (a) and (b): Latitudinal gradient of $SF_6$ (pmol mol$^{-1}$) at the surface, averaged over 2000–2010 (11 years). Panel (a) includes all longitudes in the modeled zonal average, while panel (b) includes only longitudes over the Pacific Ocean (150°E to 210°E) Modeled gradients have been scaled with respect to the South Pole. The grey symbols and their variability are calculated from the combined dataset constructed from flask data measured by the Halocarbons and other Atmospheric Trace Species (HATS) group from NOAA/ESRL and hourly CATS data (https://www.esrl.noaa.gov/gmd/hats/combined/SF6.html). Three letter codes refer to the stations (alt = Alert, Canada; brw = Pt. Barrow, Alaska, USA; mhd = Mace Head, Ireland; nwr = Niwot Ridge, Colorado, USA; kum = Cape Kumukahi, Hawaii, USA; mlo = Mauna Loa, Hawaii, USA; smo = Cape Matatula, American Samoa; cgo = Cape Grim, Tasmania, Australia; spo = South Pole). Variability is calculated as the standard deviation of monthly time series of the station data relative to the South Pole station. Panel (c): Composite of the AoA (years) of tracer "NHsurface" at the SH and tracer "SHsurface" at the NH (see main text, averaged over 2000–2010). Panel (d): AoA difference (years) calculated from the values of panel (c). The AoA at the SH of tracer "NHsurface" is subtracted from the AOA at the NH of tracer "SHsurface", and plotted against latitude.

to differences in boundary layer mixing and convection, compared to the IFS model version that was used for the ERA-Interim simulation (cy31r2).

From Figure 6 (panels (a) and (c)) it is obvious that the TOMCAT model shows the strongest $SF_6$ gradient, along with the slowest IH transport as diagnosed from the tracers "NHsurface" and "SHsurface". NIES shows smaller $SF_6$ gradients, but IH

5 AoA values in line with other models. Differences between the other models are less pronounced. The gradient of $SF_6$ is driven by emissions that take place mainly at midlatitude land masses in the NH. In contrast, tracers "NHsurface" and "SHsurface" are forced at the entire surface of both hemispheres. To make a meaningful comparison to the simulated latitudinal gradients of $SF_6$, we evaluate the AoA of tracers "NHsurface" and "SHsurface" at 50°S and 50°N, respectively, and at the surface of the models. By adding these two ages, we align the resulting composite AoA with the bulk of the calculated latitudinal $SF_6$

10 gradient between the South Pole and 50°N in panel (b) of Figure 6. The result is shown in Figure 7. In this representation, the NIES model deviates by simulating weaker latitudinal $SF_6$ gradients. TOMCAT simulates a similar $SF_6$ gradient compared to other models, but differs in the composite AoA. Both models have weak vertical mixing (see Figure 2), but NIES has fast

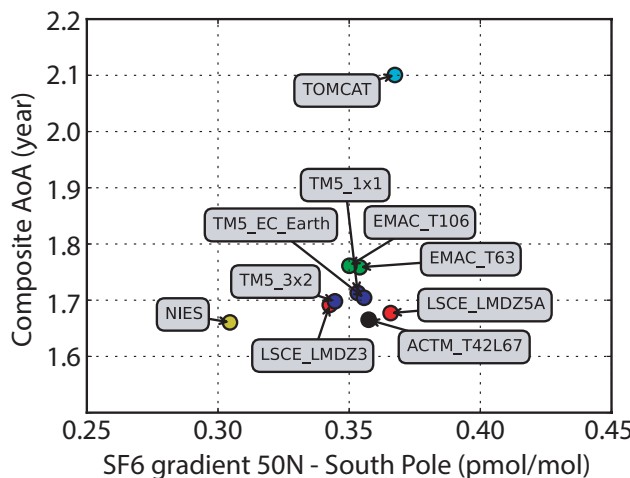

**Figure 7.** Composite AoA (years) plotted against the modelled SF$_6$ gradient. The composite AoA is calculated as the sum of the 50°N AoA of tracer "SHsurface" and the 50°S AoA of tracer "NHsurface". The SF$_6$ gradient is taken as the concentration difference between 50°N and the South Pole from the middle panel of Figure 6. Results are averaged over 2000–2010 (11 years).

horizontal transport, while TOMCAT accumulates SF$_6$ over the source regions over land (see Figure 6(a)). Different model versions cluster together in this representation, but differences between the LMDZ5A and LMDZ3 are larger. This illustrates the impact of the parameterisation of convective mixing in the LMDZ models (see Section 2.2).

### 3.5 Vertical transport

5   In this section, we focus on the vertical transport in the troposphere, as diagnosed from tracer "Land" and the simulated vertical gradients of $^{222}$Rn. To this end, we show in Figure 8 (left panel) the simulated mean vertical profiles, area weighted between 60°S and 60°N on a log-scale. Clearly, the TOMCAT and NIES models retain the $^{222}$Rn closer to the surface, because of slower vertical mixing. The other models mix the $^{222}$Rn to higher altitudes, but larger differences are found at pressures smaller than 400 hPa, where some models exhibit stronger increases at 150–300 hPa (e.g. ACTM, LMDZ5A), associated with

10   the parameterisation of convective mixing. The right-hand panel of Figure 8 shows the vertical profiles of the AoA of tracer "Land", averaged in the same manner.

    To investigate the consistency of differences in vertical mixing between the models, Figure 9 compares the simulated vertical gradient in $^{222}$Rn to the simulated AoA profiles of tracer "Land". Because $^{222}$Rn decays radioactively with a half-life of 3.8 days, we convert the vertical gradient of $^{222}$Rn mixing ratio as:

$$\Delta = \ln{}^{222}Rn(950\ \mathrm{hPa}) - \ln{}^{222}Rn(500\ \mathrm{hPa}), \tag{3}$$

    where we do not sample directly at the surface to avoid differences in sampling strategies between models (see Section 2). Here we quantify the gradient with respect to the 500 hPa level. In Figure 9 we plot the AoA gradient between 500 hPa and 950

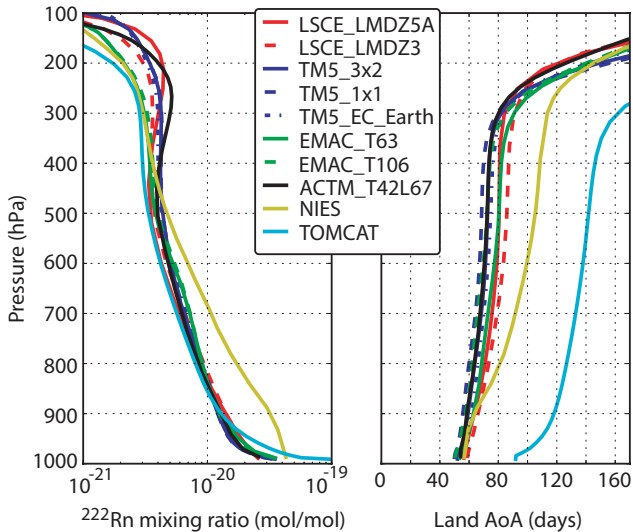

**Figure 8.** Vertical profiles of $^{222}$Rn (left, mol mol$^{-1}$, logarithmic axis) and the AoA of the "land" tracer (right, in days) simulated by the different models. The vertical profiles have been area-weighted-averaged between 60°N and 60°S and over the years 2000-2010 (11 years).

hPa against $\Delta$. The results of the different models show a high degree of linear correlation, indicating that models with efficient vertical mixing (e.g. TM5, ACTM) have small AoA differences between the boundary layer and 500 hPa along with a flatter $^{222}$Rn profile. Models with slow vertical mixing (NIES, TOMCAT) show steeper $^{222}$Rn profiles along with larger $\Delta$ values. Interestingly, the TM5 and EMAC models show that higher spatial resolution models lead to faster vertical mixing. This is different to the results found for IH transport and likely due to the fact that deep convective mixing is a sub-grid scale process. Numerics of such sub-grid scale processes are less prone to numerical diffusion compared to the advection process that drives the IH gradient. As expected, the two model versions of LMDZ that differ by their convective parameterisation show distinct differences in the vertical profiles.

## 3.6 Mapping the tropopause

Prather et al. (2011) introduced an idealised tracer (E90) to delineate the boundary between troposphere and stratosphere in transport models. Driven by surface emissions and a decay rate of 90 days$^{-1}$, tracer E90 is meant to quantify the rate of mixing of tropospheric air. Through numerical experiments, the tropopause in the UCI CTM (Prather et al., 2011) was defined as the surface on which the mixing ratio of E90 is 90 nmol mol$^{-1}$. Note that emissions of E90 are defined such that the global stationary state concentration of E90 is 100 nmol mol$^{-1}$ (see Section 2). Mixing ratios in the stratosphere are lower, due to the long transport times. As alternative for E90, the AoA tracer "Surface" may also be used to delineate the stratosphere from the troposphere, as shown in Figure 2. In order to compare the tropopause pressure calculated by either E90 or the AoA surface tracer, we plot the tropopause pressure as a function of latitude in Figure 10. The tropopause pressure is calculated based on 2000–2010 zonal averages of E90 (unit nmol mol$^{-1}$) and "Surface" AoA (unit days). For each latitude the tropopause pressure

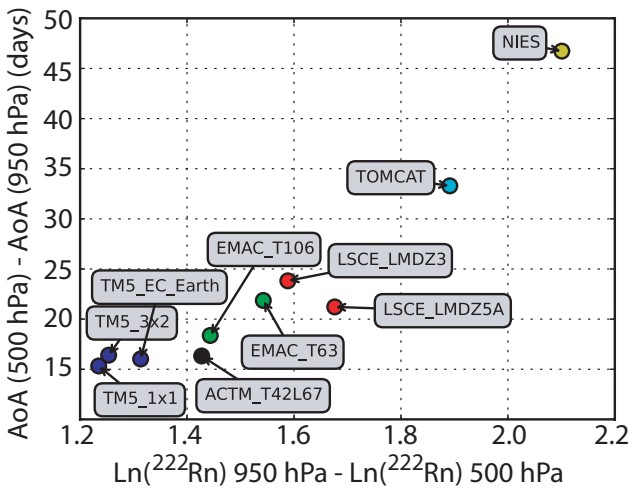

**Figure 9.** The 500 hPa - 950 hPa difference in AoA (days) of the "land" tracer plotted against $\Delta$: a measure for the vertical gradient in $^{222}$Rn (see main text). The vertical profiles used for the calculations are shown in Figure 8.

is determined by interpolation to E90 = 90 nmol mol$^{-1}$ (dotted lines in Figure 10) and to "Surface" AoA = 90 days (solid lines in Figure 10). Results show consistency for these two metrics, although larger differences occur for several models. Differences in tropopause pressure based on E90 and tracer "Surface" are likely caused by the transport characteristics of the models. The "Surface" AoA simulation is forced by linearly increasing boundary conditions which are converted to AoA, while the E90

simulation is driven by surface emissions and a decay process with a 90-day turnover time. This causes different tracer gradients and hence difference in advective and convective transport. As a result, ACTM calculates systematically higher tropopause pressures for E90, while for NIES the reverse is observed. Other models show similar estimates for both tracers. When models are compared, both tracers calculate similar tropopause pressure differences, with NIES showing a very deep tropopause at higher latitudes and TOMCAT a shallower tropopause. Off-line models that use the same driver meteorological data (e.g.

TOMCAT and TM5) would be expected to have a similar tropopause based on common temperature fields. However, since the tropopause pressures derived here depend on the transport of tracers by advection and convection, substantial differences are found that also depend on the choice of concentration (for tracer E90) and AoA (for tracer "Surface") at which the tropopause is evaluated. Finally, all models agree on a hemispheric asymmetric average tropopause with a tropopause pressure maximum around 55°S, likely associated with enhanced stratosphere–stratosphere exchange in the SH (Holton et al., 1995). In most

models, this pressure maximum is more pronounced for tracer E90.

## 4 Discussion

This TransCom AoA inter-comparison shows that the tropospheric AoA concept provides useful information on model transport characteristics. It was shown that the IH transport timescale in a particular model is strongly connected to the efficiency of

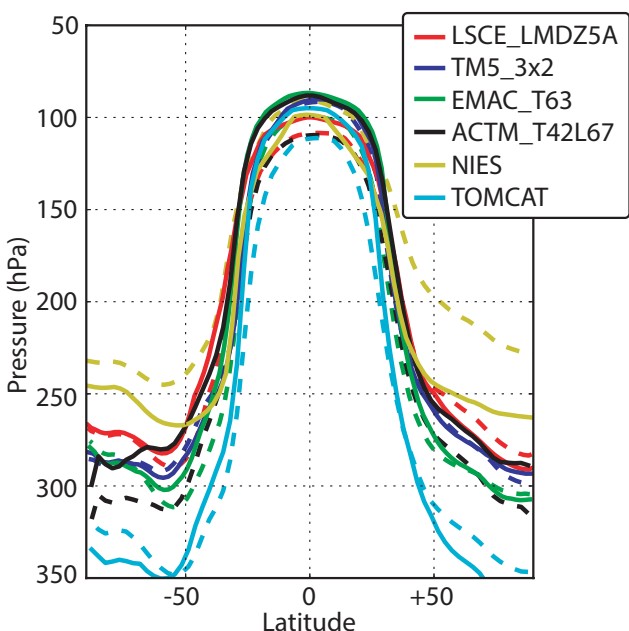

**Figure 10.** Zonal average tropopause pressure (averaged over 2000–2010) as a function of latitude, obtained from AoA tracer "surface" (solid lines) and E90 (dashed lines) for different models (see legend). Tropopause pressure is obtained from 1D linear interpolation to 90 days for AoA tracer "surface" and to 90 nmol mol$^{-1}$ for E90.

vertical mixing and hence to the specific implementation of sub-grid-scale convective transport. Thus, the AoA metric may be used to better understand flux differences derived from $CO_2$ and $CH_4$ flux inversions (Law et al., 2008; Patra et al., 2011). The NIES model, which has slow convective mixing, still shows fast IH mixing, a small IH $SF_6$ gradient, and a deep extra-tropical tropopause. In contrast, the TOMCAT model in the configuration used here combines weak vertical mixing with a stronger

$SF_6$ gradient between land and ocean, and a shallower tropopause. The TM5 model, which was diagnosed with slow IH transport in the TransCom methane inter-comparison (Patra et al., 2011), now uses the convective fluxes stored in the ECMWF ERA-Interim archive, which are based on Gregory et al. (2000) and other changes (Tsuruta et al., 2016; Dee et al., 2011). This brings the IH transport times close to other models (see Figure 7). This confirms the fact that the parameterisation of sub-grid convective fluxes in CTMs deserves attention, specifically when used in atmospheric inversion studies. This is in line with the

study of Stephens et al. (2007), who showed that the attribution of the global $CO_2$ land sink depends strongly on the vertical mixing in models.

We found an interesting hemispheric asymmetry in the IH transport times, with "older" NH air diagnosed with an AoA tracer forced at the SH surface, compared to the age of SH air, diagnosed with an AoA tracer forced at the NH surface. We tentatively attribute this asymmetry to the effect of land-masses in the NH, which lead to an atmospheric stabilisation in winter,

decoupling the BL from the overlying free troposphere (FT). In the AoA protocol, this affects the surface forcing. The situation is reversed in summer, with more efficient mixing over land-masses. The resulting net effect is well known as the "seasonal

rectifier" (Denning et al., 1995). In Appendix B we present a simplified numerical experiment that shows that the mean AoA in the upper atmosphere shortens with a larger seasonal cycle in near-surface mixing, in line with the asymmetry in IH transport as found in all models. Thus, the AoA metric can also be used to quantify the strength of these rectifier effects in CTMs. Another (related) factor that might be responsible for this asymmetry is the mean position of the Inter Tropical Convergence

Zone (ITCZ) north of the equator (Schneider et al., 2014).

A further important timescale in GCMs and CTMs is the mixing time of the BL and its coupling with the overlying FT. Together with the aforementioned "convection" timescale, these two quantities appear responsible for the main model diversity in this inter-comparison. For instance, the TM5 model shows a relatively strong coupling between the BL and the FT, witnessed by (i) relatively young air of the ocean AoA tracer over land (Figure 4), (ii) only a small IH difference in the AoA (Figure 6d),

and (iii) a small 950 hPa – 500 hPa vertical gradient in $^{222}$Rn and the land AoA tracer (Figure 9). Differences with the EMAC, LMDZ, and ACTM models seem to be determined by the strength of this BL–FT coupling. Differences with NIES and TOMCAT, however, seem to be driven by differences in the convective parameterisation.

Other model differences, such as resolution, advection scheme, the meteorological driver data, nudging, and the vertical coordinate system may also play a role and may become more apparent on the smaller spatial and temporal scales that are not

part of this first analysis. In mapping the tropopause with either E90 or the AoA derived from the surface tracer (Figure 10) model-dependent differences appeared that may be related to the treatment of atmospheric transport. AoA tracers with a linearly growing boundary condition lead to different concentration gradients than tracers with surface emissions and first-order decay, such as E90. In regions with large gradients, such as the planetary boundary layer and the tropopause, differences in the numerical treatment of advection may lead to different transport characteristic of AoA tracers, compared to tracers with

physical sources and sinks.

Our multi-model results agree qualitatively with the study of Waugh et al. (2013) that focused on SF$_6$ and a AoA tracer in a single model (GMI-MERRA). They found "older" AoA than the age derived from SF$_6$ observations. The AoA simulations presented here allow for more detailed analyses in the future, including comparisons to earlier efforts to quantify tropospheric AoA based on observations (e.g., Holzer and Waugh, 2015).

The simple AoA concept exploited here is easily implemented in CTMs and GCMs, and other modellers are encouraged to implement AoA tracers in their models. The AoA protocol is also useful as benchmark for model development, similar to Prather et al. (2008). Analysis of AoA simulations performed with a single model with different advection schemes, convective parameterisations, or nudging schemes can be used to study their impact on large scale transport features, such as IH and vertical transport. AoA studies with a single transport model and different meteorological driver data (e.g. ERA-Interim versus

JRA-25) would reveal the potential role of the meteorological driver data on AoA biases. Finally, the entire time series (1988–2015) may be used to study trends in the Brewer Dobson circulation (Fu et al., 2015) or IH transport timescales.

## 5 Conclusions

This paper presents the first results of the TransCom AoA inter-comparison. Six models simulated five AoA tracers and four additional tracers over the time period 1988–2015. AoA tracers were forced by linearly growing boundary conditions in a predefined atmospheric volumes. Advantages of this approach are that (i) the forcing volume can be flexibly chosen, and (ii) transport time-scales can easily be derived based on simulated mixing ratios that indicate when an air mass was "last in contact" with the boundary. Disadvantages are that (i) there are no known atmospheric species with linear growing boundary conditions, and (ii) artificial mixing ratio gradients introduced close to the forcing volume may be challenging for advection schemes. Successful implementation of the protocol in six global models revealed interesting differences in large-scale transport features. In this paper we mainly analysed averages over the 2000–2010 period. The main findings of this study are:

1. The inter-hemispheric transport time depends strongly on the strength of the convective parameterisation in the participating models. Although convective mixing is identified as a major cause for inter-model differences, other causes, such as the source of reanalysis data, nudging, and advection scheme, cannot be ruled out at the moment. It is recommended to apply the AoA protocol in a single model with different set-ups to study the impact on large-scale transport features.

2. Inter-hemispheric transport proceeds faster from the NH to the SH than from the SH to the NH. This is attributed to the seasonal rectifier effect caused by NH landmasses, with strong mixing in summer and weak mixing in winter as shown in Appendix B.

3. Boundary layer mixing and venting to the free troposphere over land varies among models, which leads to consistent differences in modeled vertical gradients. The TM5 model shows fast vertical mixing, with consequently relatively young air of the ocean AoA tracer over land (Figure 4). In contrast, the TOMCAT model shows slow vertical mixing, and old air of the ocean AoA tracer over land.

4. The AoA concept can be used to map the tropopause. At the tropopause the air has last been in contact with the surface 90 days ago (AoA = 90 days). For most models, the derived tropopause pressure is in good agreement with the E90 tracer (Prather et al., 2011).

5. Upper stratospheric AoA ranges considerably among the participating models (4 –7 years).

In general, further analysis is required to fully exploit the simulations presented in this paper. For instance, the analysis of inter-annual variations in inter-hemispheric transport and stratospheric AoA may reveal interesting differences between the models. Most of all, further analysis should focus on the causes of the still large spread in the participating models.

## 6 Code availability

The AoA protocol and analysis software in the form of a Jupyter Notebook (python) are available to the community through github: https://github.com/maartenkrol/AoA. The python notebook *aoa_paper.ipynb* needs *AoA_tools.py*. The model output,

converted to a standard format, can be downloaded from the ftp location mentioned in *AoAprotocol.pdf*. Apart from the AoA tracers discussed in this paper, also AoA tracers are available that are forced in the troposphere and stratosphere. These tracers are forced above and below a climatological pressure surface given by Lawrence et al. (2001). Results are not discussed, because not all models used the correct definition. Information on the availability of source code for the models featured in this

paper is tabulated in Table A1.

## Appendix A:  Comparison with HIPPO SF$_6$

Figure A1 compares monthly mean SF$_6$ output from the models to observations made during the High-performance Instru-mented Airborne Platform for Environmental Research (HIAPER) Pole-to-Pole Observation (HIPPO) campaigns (Wofsy et al. , 2011). Only HIPPO flights over the Pacific Ocean are included, and the corresponding model mixing ratios are averaged

over the Pacific Ocean only (from 150°E to 210°E). Hippo 1 observations are compared to monthly means of January 2009, Hippo 2 to November 2009, Hippo 4 to June and July 2011 averages, and Hippo 5 to August 2011. The left panels show the latitudinal gradients averaged over 1–3 km. Since models used different SF$_6$ 1988 mixing ratios as initial fields and show different accumulation rates in the lower atmosphere, models have been shifted to match the 2009 observations. Note that the shifts (mentioned in the caption) range from +1.3 ppt for TOMCAT to –0.4 ppt for NIES. During the 2009–2011 Hippo period

differences between these two models gradually increase further. This is likely related to fast accumulation in NIES due to limited vertical mixing, combined with fast horizontal transport in the lower atmosphere. TOMCAT was also diagnosed with slow vertical transport (see Figure 2) but also with slow horizontal mixing (signalled by "old" air over the ocean for AoA tracer "Land" in Figure 4). Other models agree on a faster increase of SF$_6$ in the models compared to observations, signalling a likely overestimate of the emissions in 2009–2011 (see Table 3). The latitudinal gradients are very similar among the models

(except for NIES as in Figure 6) and generally agrees well with observations. The right-hand panels show the modeled and measured vertical SF$_6$ gradients, calculated by taking the difference between 5–7 km averages and 1–3 km averages, as in Patra et al. (2014). Modeled and measured vertical gradients are very small and measurement noise hampers us from drawing firm conclusions about the ability of models to correctly simulate SF$_6$ vertical gradients.

## Appendix B:  Seasonal rectifier effect

The intention of this Appendix is to calculate the effect of seasonal vertical mixing over land on the calculated mean AoA in the upper atmosphere. In the main manuscript it is speculated that a seasonal rectifier effect partly explains why the AoA derived from "SHsurface" around the North Pole in the models is 0.10 – 0.17 years older than AoA derived from "NHsurface" around the South Pole (Figure 6(d)). With more land cover, seasonality in mixing is stronger on the NH compared to the SH. To illustrate the seasonal rectifier effect, we implemented the AoA protocol in a simplified three box model. To this end, we

force a surface box of 100 m (pressure difference $\Delta p_1$ between bottom and top = 13 hPa) with a linearly growing mixing ratio as boundary condition. The surface box mixes with an overlying boundary layer box ($\Delta p_2$ = 137 hPa). This boundary layer

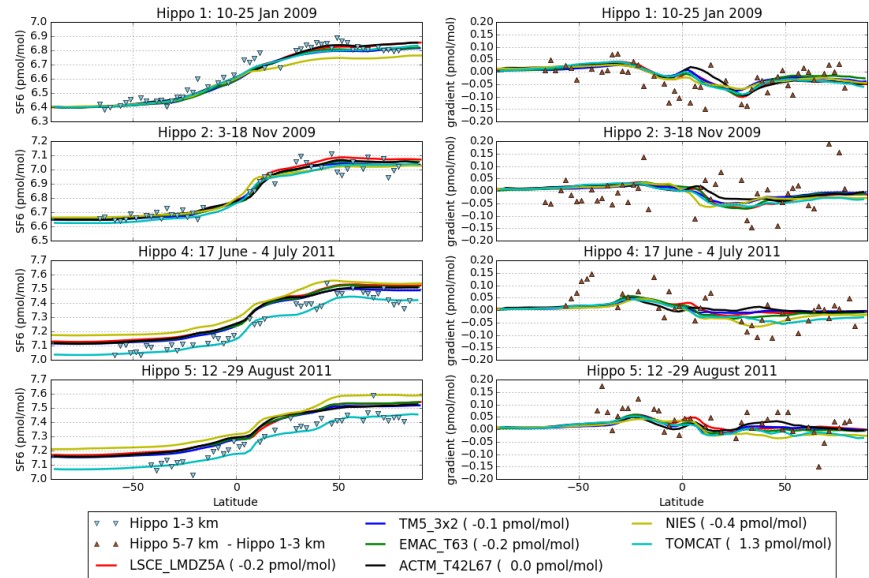

**Figure A1.** Comparison of modeled monthly-mean SF$_6$ (averaged over the Pacific Ocean only: from 150°E to 210°E) to the High-performance Instrumented Airborne Platform for Environmental Research (HIAPER) Pole-to-Pole Observation (HIPPO) campaigns (Wofsy et al. , 2011), similar to comparisons presented in Patra et al. (2014). Left panels show models and observations averaged over 1-3 km altitude. Right panels are the gradient between 5-7 km averages and 1-3 km averages. Models have been shifted to match the HIPPO latitudinal gradients in 2009. Magnitudes of the shifts are indicated in the caption.

box additionally mixes with the overlying free atmosphere ($\Delta p_3$ = 600 hPa). Mixing timescales between the boundary layer and the forced surface layer ($\tau_1$), and between the boundary layer and free atmosphere ($\tau_2$) vary with season according to:

$$\tau_1 = (a_1 \sin(2\pi t) + 1.0)\,\frac{1}{365} \tag{B1}$$

$$\tau_2 = (a_2 \sin(2\pi t) + 1.0)\,\frac{7}{365} \tag{B2}$$

5    where time $t$ is measured in years, and $a_1$ and $a_2$ are the amplitudes of the seasonal cycles imposed on the standard mixing timescales $\tau_1$ and $\tau_2$ of 1 and 7 days, respectively. The following set of differential equations is solved to simulate the mixing ratios $x_1$, $x_2$, and $x_3$ according to the AoA protocol:

$$\frac{\mathrm{d}x_1}{\mathrm{d}t} = 1$$

$$\frac{\mathrm{d}x_2}{\mathrm{d}t} = \frac{x_1 - x_2}{\tau_1} - \frac{x_2 - x_3}{\tau_2}$$

10    $$\frac{\mathrm{d}x_3}{\mathrm{d}t} = \frac{\Delta p_2}{\Delta p_3}\frac{x_2 - x_3}{\tau_2}$$

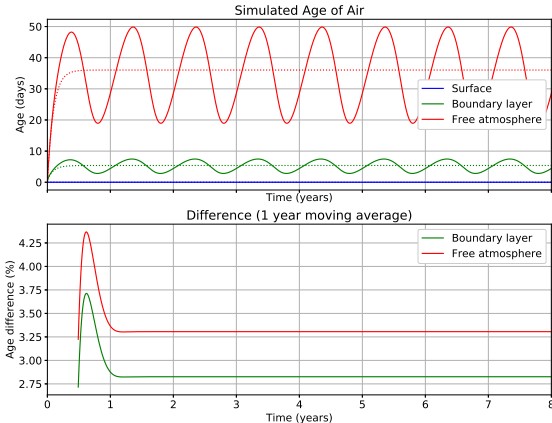

**Figure B1.** Upper panel: Simulated AoA in the simple three-box model described in the main text. The solid lines show the NH simulation with a 50% seasonal cycle in vertical mixing. The dotted lines show the SH simulation with no seasonal cycle in vertical mixing. Lower panel: Difference in the one-year moving average AoA calculated as $100 \times \frac{\text{SH}-\text{NH}}{\text{NH}}$.

where $\frac{\Delta p_2}{\Delta p_3}$ accounts for the fact that boxes 2 and 3 have different pressure thicknesses. We now performed two simulations, one representing the SH with no seasonal cycle in mixing ($a_1 = a_2 = 0$), and one representing the NH, with a 50% amplitude in seasonal mixing ($a_1 = a_2 = 0.5$). Results of the simulations, with mixing ratios converted into AoA units ($AoA = t - x$), are plotted in the upper panel of Figure B1. As expected, NH AoA (solid lines) shows a seasonal cycle in the boundary layer

and free atmosphere, while the SH simulation (dotted lines) reaches a steady-state AoA in the free atmosphere of about 37 days, roughly in line with the multi-year averaged AoA of tracer "Surface" in Figure 2. The lower panel of Figure B1 shows the percentage difference in the one-year moving average of the NH and SH simulations, calculated as $100 \times \frac{\text{SH}-\text{NH}}{\text{NH}}$. Indeed, the average AoA in the free atmosphere and boundary layer is younger by about 3% when a seasonal cycle is applied to the vertical mixing. Thus, even when the mean mixing timescales between the boxes are the same, the resulting mean AoA in the

upper atmosphere is different. In a real atmosphere, other factors may also play a role, such as convection and the asymmetry in the mean position of the ITCZ north of the equator (Schneider et al., 2014). Nevertheless, this simple example shows that a rectifier effect may partly explain the asymmetry in inter-hemispheric mixing as seen in all models (Figure 3).

*Author contributions.* M.K. wrote the manuscript and incorporated revisions and comments from all co-authors. M.d.B performed the TM5 simulations, L.K. the TM5_EC-Earth simulations. H.O. and A.P. were responsible for EMAC, Y.Y., F.C., and P.B. for LMDZ, P.P for ACTM,

D.B. and S.M. for NIES, and S.D, W.F, and M.P.C for TOMCAT. The protocol was initiated by M.K. and P.P.

**Table A1.** Availability of source code for the models featured in this paper.

| Short Name | Code availability |
|---|---|
| LMDZ | LMDZ is open source free software under licence CeCILL (http://www.cecill.info/licences/Licence_CeCILL_V2-en.html). The versions used here are available online via http://svn.lmd.jussieu.fr/LMDZ. |
| TM5 | TM5 version control is performed on the SVN server of the Dutch Met Office (KNMI). The AoA simulations are performed using TM5-MP, revision r182, committed on 29-05-2017. Access to the SVN server is granted to researchers actively participating in model development. |
| TM5_EC-Earth | TM5_EC-Earth is the TM5-MP version incorporated in EC-Earth version 3.2.2. Information about the model and access to the code is available at http://www.ec-earth.org. The specific version used for this study can be found under the branch r4353-Age_of_Air. This version consists of the main developmental version of EC-Earth updated until 04-07-2017, supplemented with the AoA code of TM5. |
| EMAC | The ECHAM/MESSy Atmospheric Chemistry (EMAC) model is a configuration of MESSy, which is being continuously further developed and applied by a consortium of institutions. The usage of MESSy and access to the source code is licensed to all affiliates of institutions that are members of the MESSy Consortium. Institutions can be a member of the MESSy Consortium by signing the Memorandum of Understanding. More information can be found on the MESSy Consortium website (http://www.messy-interface.org). |
| ACTM | The ACTM code is not open source. The model code is based on CCSR/NIES/FRCGC AGCM5.7b. The copyright belongs to the developers at CCSR (Univ. Tokyo), NIES and FRCGC (JAMSTEC). |
| NIES | The NIES source code is proprietary. The model copyright is owned by developers at NIES. |
| TOMCAT | The TOMCAT code is not open source. The model is available to all UK researchers funded by the Natural Environment Research Council (NERC) and other UK research councils. The model is also available to other scientists with an interest in active collaboration with existing users, subject to limitations of resources to support the collaborations. |

*Competing interests.* The authors declare that they have no conflict of interest.

*Acknowledgements.* We thank the HIPPO science team and the crew and support staff at the NCAR Research Aviation Facility, and all the laboratory staff working for AGAGE and NOAA measurement networks. The TM5 model simulations and analysis was carried out on the Dutch national e-infrastructure with the support of SURF Cooperative. Marco de Bruine is supported by the Netherlands Organization for Scientific Research (NWO), project number GO/13-01. The TOMCAT simulations were performed on the UK Archer and Leeds ARC HPC facilities. Dmitry Belikov is supported by the Japan Society for Promotion of Science, Grant-in-Aid for Scientific Research (S) 26220101. Maarten Krol is supported by ERC AdG grant 742798. Paul Palmer is acknowledged for support of the initial AoA initiative.

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
