# Peer review of "Age of air as a diagnostic for transport time-scales in global models"

_Geoscientific Model Development, 2017_

## Referee Comment (RC1) · Anonymous Referee #1 · 6 Dec 2017

The manuscript presents a diagnostic for inter-hemispheric transport in global models. The method is very useful to analyse model's transport bias and to link these biases with the model parameters (convection, resolution, data). The results compare very well with observations.

Although the paper contains some interesting material, which should be published, the manuscript itself could be improved qualitatively in some parts. Some paragraphs and sections need revisions by enhancing the discussion with regard to the scientific content as well as to the models differences (used resolution, convection schemes, data). Particularly, the difference in the reanalyses (JRA-25 and ERAI) could be emphasize. The work should be published after major revisions. In the following here are my major points and general concerns:

**Major points:**

1. *Mainly, I am missing a critical discussion and the connection between the model's difference with the different set-up, such as vertical and horizontal resolution, nudging and reanalyses-driven models. How different configurations in Table 4 affect the results. Convective parameterisation may play the biggest role but one should not neglect the differences induced by the difference in the reanalyses, vertical resolution, nudging.*

2. *My other major point concerns the organisation of the figures. Some of them could be grouped to allow an easy reading and inter-comparision. I would suggest to regroup on same panel figures 3 and 4; fig. 5 and 6; Fig. 8 and 9.*

3. *Section 5 (Conclusions) provides mostly a brief summary of things which have been stated before.*

Minor points:

1. *Page 2, line 12: ". . . is only broken down in the upper stratosphere . . ." Please add citation after "stratosphere" about stratospheric SF6 photo-dissociation*

2. *Page 2, line 32: ". . . upper-tropospheric equatorial westerly duct,. . ." Please cite Waugh and Funatsu, 2003 after "duct"*

3. *Page 3, line 1: please replace "again" by "also"*

4. *Page 3, lines 9: ". . . troposphere-stratosphere exchange . . ." please cite Holton et al., 1995 which is the suitable reference for STE*

[Figure]

5. ***Page 3, line 12: "Stratospheric age of air and its temporal trend have been determined from SF6 measurements from the MIPAS satellite (Stiller et al., 2012)" and from balloon observations (Engel et al, 2009).***

6. ***Page 5, line 21-23: The tracers that are not used in this study don't need to mentioned or listed in this paper. Please remove this sentence. "Note that we also included a 222 Rn simulation with monthly varying emissions over Europe during 2006–2010, based on the high resolution emissions maps presented in Karstens et al. (2015). The current paper will, however, not analyse these simulations."***

7. ***Page 9, lines 2: " For this inter-comparison . . ." coma after "inter-comparison"***

8. ***Page 9, lines 5-7: "Recent analysis (Tsuruta et al., 2016) shows that the mass fluxes produced with the ERA-interim data set (Dee et al., 2011) lead to faster inter-hemispheric transport compared to the old model version using the Tiedtke (1989) scheme that was used in the earlier TransCom study (Patra et al., 2011)." Sentence should be revised "According to Tsuruta et al., 2016, the mass fluxes . . . ". There is also an over citing Dee et al., 2011 in this page.***

9. ***Page 10, line 28: ". . . (Austin and Houze Jr, 1973; Belikov et al., 2013a) A modified . . . " there is a missing dot before the "A . . ."***

10. ***Page 12, line 7: ". . . signalling stratosphere-troposphere exchange. " Please add Holton et al., 1995 at the end of the sentence.***

11. ***Page 12-13, line 12/line 1: " Here, all models agree on an interesting asymmetry: AoA derived from "SHsurface" around the North Pole is older than AoA derived from "NHsurface" around the South Pole." Why?***

12. *Page 13, line 1: What the cause of the differences between TOMCAT and NIES results?*

13. *Page 13, line 12: ". . . emission from the NH." Please replace by ". . . from the NH emissions."*

14. *Page 14, line 1: "ndeed, the lowest CH4 concentrations on the Earth's surface are found at NOAA site Eastern Island (EIC) (Patra et al., 2009b), which was attributed to "old" air in combination with strong removal of CH4 by OH at tropical latitudes". Please rephrase this sentence.*

15. *Page 14, line 6: How Louis (1979) scheme would impact the transport? Please be more explicit.*

16. *Page 14, line 10: "Here it should be noted that the TOMCAT and NIES AoA is already systematically older at the tropopause (see Figure 2)." But in the stratosphere, the AoA from LMDZ, TM5, EMAC are even older than TOMCAT and NIES. Therefore, the sentence is not useful here.*

17. *Page 15, line 9: "Further analysis on the stratospheric AoA in this model ensemble is left for future exploration." Before this sentence please discuss Garny et al., 2014; Ploeger et al, 2015 concerning the impact of the aging by mixing which play important role.*

18. *Page 17, paragraph 2, line 5-15: Please combine fig. 8 and 9 and rephrase the paragraph. This will make this paragraph easy to follow. Please improve the ticks and legend of actual fig. 8. Enumerating the panel would be helpful.*

19. *Page 23, line 21-22: Please rephase this ". . . five AoA tracers..."*

**References**

Holton, J. R. and Haynes, Peter H. and McIntyre, M. E. and Douglass, A. R. and Rood, R. B. and Pfister, L. (1995), Stratosphere-troposphere exchange, *rev. Geophys.* , *33*,403-440, doi:10.1029/95RG02097.

Engel, A., T. Möbius, H. Bönisch, U. Schmidt, R. Heinz, I. Levin, E. Atlas, S. Aoki, T. Nakazawa, S. Sugawara, F. Moore, D. Hurst, J. Elkins, S. Schauffler, A. Andrews, and K. Boering (2009), Age of stratospheric air unchanged within uncertainties over the past 30 years, *Nature Geoscience*, *2*, 28–31, doi:10.1038/ngeo388.

Garny, H., T. Birner, H. Bönisch, and F. Bunzel (2014), The effects of mixing on age of air, *J. Geophys. Res. Atmos.*, *119*, doi:10.1002/2013JD021417.

Ploeger, F., M. Abalos, T. Birner, P. Konopka, B. Legras, R. Müller, and M. Riese (2015b), Quantifying the effects of mixing and residual circulation on trends of stratospheric mean age of air, *Geophys. Res. Lett.*, *42*, doi:10.1002/2014GL062927.
* * *

---

## Referee Comment (RC2) · M.J. Prather (Referee) · 8 Jan 2018

gmd-2017-262 A new look at patterns of inter-hemispheric mixing and how the troposphere disperses pollutants.

This is a clearly written paper, easy to read, that describes the results from a TransCom type of study in which a group of models ran the same protocol of trace species in order to test large-scale transport and mixing. There a fundamental difficulty with the protocol itself, and it should be revised before the next generation of studies in inter-hemispheric transport are begun. Nevertheless, it is after the model runs have been completed, and I would just recommend that the authors admit the inconsistency and make the most of what has been done. There is still considerable science to be gotten

out of the results, but one should be more careful about drawing conclusions when comparing the fundamentally different protocols: fixed surface abundances vs. fixed surface fluxes.

Protocol Problem: The age-of-air (AoA) tracers are forced with surface abundances while all the others (SF6, Rn, e90) are flux driven. The problem with surface abundances is (1) the flux from surface to atmosphere will depend on each model, particularly seasonal mixing, and (2) there is no atmospheric species that I know of with a fixed surface abundance. Notice that e90, while an artificial tracer, is still flux driven. This protocol makes it extremely difficult to compare the AoA lessons with the SF6 or Rn observations and draw any conclusions about transport. For example, SF6 and Rn have absolutely no "diode/rectifier effect" since the emissions cannot be stopped and must get out of the boundary layer daily. The AoA tracers can indeed have a diode effect. Alternative approached to defining AoA can be easily defined with the flux boundary – see the CO2 example in Prather et al., 2008.

P3-L17-24. I confess to working in this topic of hemispheric exchange and AoA for 3 decades. I find the referencing OK, but missing some of the original work. Possibly the first stratospheric modeling study to examine age of air and define it for CO2 is Hall and Prather (1993). The Prather et al (2008) work is one of the most thorough efforts to test different advection schemes with the identical wind fields. There are also some very interesting results in Hsu 2014 showing that even the cycle version of the ECMWF fields can notably change STE fluxes.

P7 - are the monthly mean mixing ratios 3D? please state if so. Likewise are the hourly station data 1D (profiles) or just layer 1.

P8 L14 – do you mean 'adapted' instead of adopted?

P11 L29 – is 'IH' defined earlier – might be better just spelling it out everywhere, after all you did not define VT for vertical transport?

[Figure]

P12 L13 – please give real numbers - -how many years older?

P13 L1-4 – The transport from N-to-S in the upper troposphere was clearly shown in the first 3D CFC simulations (Prather et al., 1987), what is truly new here is the S-to-N transport patterns, which have not yet been so well documented, e.g., Fig 3 vs Fig 4. Can you explain why the mechanism is different? Looking at diagnostics of cross-equatorial transport in our model, I suspect that the S-to-N transport occurs at lower altitude via the Indian Ocean monsoons.

P13 L8 – The 'oldest' tropospheric air noted here was also part of the Prather e90 paper, at least compare.

P14 L1 – "Easter Island"?

P14 L7 – This conclusion about BL causing the differences seems speculative. Could you test?

P15 L7 – In addition to Diallo et al 2012, check the Prather 2008 PNAS paper that demonstrated the convergence of two schemes with increased vertical resolution.

P16 L7 – Remember that the SF6 is flux-driven and may show different features than the AoA tracers.

P18 L1 – It really does not matter that TOMCAT accumulates SF6 in the BL because the flux out is the same. The most you would "accumulate" is 1 day's emissions. Try another explanation.

P19 – I like figures 9 and 10. But what is with the NIES result" should not these all go to 0 at the equator? Figure 10 is fun, but not sure how to read it. TOMCAT does not look anomalous in Fig 9 but does in Fig 10? Am I missing something here?

P23 – Please spell out IH and BL etc in the conclusions. The language here should stand alone and not require memory from the rest of the paper.

The Conclusions are a bit too speculative. I think they need to be re-thought and toned

down. This section is the one requiring the most work.

e.g., We know that the models all have different convection, but even if running the same meteorology there are so many other differences (advection scheme, mass conservation scheme, . . .) that to say that the convective parameterization is the primary reason could only be justified if one model ran all the different convective schemes in a consistent framework.

e.g., I don't know what a 'seasonal rectifier' is. In the long run all the excess flux from the NH must get into the SH hemisphere and vice-versa. None of it disappears (as would be the case with a rectifier). The flux N-S and S-N is the same, right? So 'faster' is not an obvious attribute. The gradients are clearly different as documented here, but faster is not a useful term.

some references:

Hall, T.M., and M.J. Prather, 1993: Simulations of the trend and annual cycle in stratospheric CO2. J. Geophys. Res., 98, 10573-10581, doi:10.1029/93JD00325.

Prather, M., M. McElroy, S. Wofsy, G. Russell, and D. Rind, 1987: Chemistry of the global troposphere: Fluorocarbons as tracers of air motion. J. Geophys. Res., 92, 6579-6613, doi:10.1029/JD092iD06p06579.

Hsu, J. M.J. Prather (2014) Is the vertical residual velocity a good proxy for stratosphere-troposphere exchange of ozone? Geophys. Res. Lett., 41, doi:10.1029/2014GL061994

Prather M.J., X. Zhu, S.E. Strahan, S.D. Steenrod, J.M. Rodriguez (2008), Quantifying errors in trace species transport modeling, Proc. Nat. Acad. Sci. 105(50): 19617-19621.

---

## Author Comment (AC1) · 5 Feb 2018

**Reply to reviewer 1**

Maarten Krol et al.

February 4, 2018

**1 Reply**

First of all we would like the thank the reviewer for his/her comments, and careful reading of the manuscript. We will follow most of the suggestions made in the revised manuscript. There are several main points made, which will be addressed as indicated below.

- *I am missing a critical discussion and the connection between the model's difference with the different set-up, such as vertical and horizontal resolution, nudging and reanalyses-driven models.* This point is also raised by Reviewer 2. The main aim of this first paper is to present the protocol and first results. The intercomparison brings a lot of interesting information, however, analysing the exact causes for the inter-model differences would lengthen the paper considerably. We agree with the fact, at the current state, it is too early to diagnose convection as the main source of the differences (see also Reviewer 2). We will rephrase out conclusions accordingly. To make a detailed analysis, models should be run with different convective parameterisations, resolution, and advection scheme. This is partly done by the LMDZ and TM5 models. To analyse the differences in nudging data (ERA-interim vs. JRA25) would require detailed analysis of the driving meteorological fields, and maybe a single model using both data streams. At this stage, we think this is beyond the scope of the paper, which presents the protocol and first analysis. This is fully in line with the scope of the GMD journal, and hopefully many follow-on studies will further analyse the results that are already available.

- *The organisation of the figures*: We will consider re-organising the figures in the revised manuscript.

- *Section 5 (Conclusions) provides mostly a brief summary of things which have been stated before.* We think this is an efficient way to present the main conclusions. We will be more textual in the revised manuscript.

Below we address the other points of the reviewer.

1. *Page 2, line 12: " is only broken down in the upper stratosphere ". Please add citation after "stratosphere" about stratospheric SF6 photodissociation.* Will be included.

2. *Page 2, line 32: " . . upper-tropospheric equatorial westerly duct,. . .". Please cite Waugh and Funatsu, 2003 after "duct".* Will be done.

3. *Page 3, line 1: please replace "again" by "also".* OK.

4. *Page 3, lines 9: ". . . troposphere-stratosphere exchange . . ." please cite Holton et al., 1995 which is the suitable reference for STE.* OK

5. *Page 3, line 12: "Stratospheric age of air and its temporal trend have been determined from SF6 measurements from the MIPAS satellite (Stiller et al., 2012)" and from balloon observations (Engel et al, 2009).* OK.

6. *Page 5, line 21-23: The tracers that are not used in this study don't need to mentioned or listed in this paper. Please remove this sentence. "Note that we also included a 222 Rn simulation with monthly varying emissions over Europe during 2006–2010, based on the high resolution emissions maps presented in Karstens et al. (2015). The current paper will, however, not analyse these simulations."* On this point, we disagree. This paper is intended as a first presentation of the intercomparison exercise, and readers should be made aware of the data that is available for further analysis.

7. *Page 9, lines 2: "For this inter-comparison . . ." coma after "inter-comparison".* OK.

8. *Page 9, lines 5-7: "Recent analysis (Tsuruta et al., 2016) shows that the mass fluxes produced with the ERA-interim data set (Dee et al., 2011) lead to faster inter-hemispheric transport compared to the old model version using the Tiedtke (1989) scheme that was used in the earlier TransCom study (Patra et al., 2011)." Sentence should be revised "According to Tsuruta et al., 2016, the mass fluxes . . . ". There is also an over citing Dee et al., 2011 in this page.* Will be revised.

9. *Page 10, line 28: . . . (Austin and Houze Jr, 1973; Belikov et al., 2013a) A modified. . . there is a missing dot before the "A.. ".* OK.

10. *Page 12, line 7: ". . . signalling stratosphere-troposphere exchange." Please add Holton et al., 1995 at the end of the sentence.* OK.

11. *Page 12-13, line 12/line 1: "Here, all models agree on an interesting asymmetry: AoA derived from "SHsurface" around the North Pole is older than AoA derived from "NHsurface" around the South Pole." Why?* This is explained in the answer to Reviewer 2. This is likely due to a seasonal rectifier effect, with wintertime trapping and efficient mixing in summer over NH landmasses. In the manuscript (abstract

and page 22), we also mention that the asymmetric position of the ITCZ may play a role. In the revised manuscript we aim to add an Appendix in which the seasonal rectifier effect will be further explained.

12. *Page 13, line 1: What the cause of the differences between TOMCAT and NIES results?* We think this issue is clearly discussed in the remainder of the manuscript, when SF6 and 222Rn results are analysed. The bottomline is that both TOMCAT and NIES are diagnosed with slow vertical transport. NIES still appears to have fast interhemispheric transport, likely related to a diffusive advection scheme.

13. *Page 13, line 12: ". . . emission from the NH." Please replace by ". . . from the NH emissions."* OK.

14. *Page 14, line 1: "Indeed, the lowest CH4 concentrations on the Earth's surface are found at NOAA site Eastern Island (EIC) (Patra et al., 2009b), which was attributed to "old" air in combination with strong removal of CH4 by OH at tropical latitudes". Please rephrase this sentence.* Will be rephrased.

15. *Page 14, line 6: How Louis (1979) scheme would impact the transport? Please be more explicit.* We will refer here to earlier studies, e.g. Chipperfield (2006) and Wang (1999) and Section 2.7, where is stated that local BL transport schemes like Louis (1979) lead to slower exchange between the BL and free troposphere.

16. *Page 14, line 10: "Here it should be noted that the TOMCAT and NIES AoA is already systematically older at the tropopause (see Figure 2)." But in the stratosphere, the AoA from LMDZ, TM5, EMAC are even older than TOMCAT and NIES. Therefore, the sentence is not useful here.* We think it is still useful to remark here that the AoA of the surface tracer differs in TOMCAT and NIES. We will slightly rephrase.

17. *Page 15, line 9: "Further analysis on the stratospheric AoA in this model ensemble is left for future exploration." Before this sentence please discuss Garny et al., 2014; Ploeger et al, 2015 concerning the impact of the aging by mixing which play important role.* OK.

18. *Page 17, paragraph 2, line 5-15: Please combine fig. 8 and 9 and rephrase the paragraph. This will make this paragraph easy to follow. Please improve the ticks and legend of actual fig. 8. Enumerating the panel would be helpful.* We will revise the figures.

19. *Page 23, line 21-22: Please rephrase this ". . . five AoA tracers..."* OK.

**2    Reference**

Wang, K. Y., Pyle, J. A., Sanderson, M. G., & Bridgeman, C. (1999). Implementation of a convective atmospheric boundary layer scheme in a tropospheric chemistry transport model. Journal of Geophysical Research, 23729–23745.

---

## Author Comment (AC2) · 5 Feb 2018

**Reply to reviewer 2. M. Prather**

Maarten Krol at al.

February 5, 2018

First of all we would like the thank the reviewer for his insightful comments, and starting this discussion. The reviewer makes one big point, and several other points that we address below.

- *The protocol has a fundamental difficulty: for the Age of Air (AoA) tracers with fixed boundary conditions are used, which cannot be compared to real tracers like $SF_6$, $^{222}Rn$, and e90.*: In order to derive transport times from transport model simulations, there are several options. In all cases a time scale has to be extracted from simulations, as time is not a transportable quantity. One option is to have tracers with different lifetimes (e.g. 222Rn, e90, ..) and fixed emissions. Alternatively, tracers with 'known' emissions and a source-sink imbalance can be used to investigate the age of air masses (e.g. SF6, CO2). In all cases, time-scales of mixing have to be derived from known decay rates, or from atmospheric accumulation rates. Some tracers have the clear advantage that atmospheric observations are available (SF6, CO2, 222Rn). This protocol is experimenting with boundary conditions with a prescribed growth rate, with the growth rate serving as the 'clock'. This offers advantages, because the boundary conditions can be applied in a flexible way (NH, SH, land, ocean, ...). As the reviewer rightfully remarks, we also face challenges. However, according to us there is not a fundamental difficulty with the protocol, other than the issues we discuss in the manuscript. Yes, we can have a rectifier/diode effect, but studying this in different models may help diagnose model–model differences. For instance, our surface boundary conditions are applied only in the lowest 100 m. This introduces a sensitivity for boundary layer mixing. Models with efficient BL mixing (like TM5) consequently find 'young air' in the upper boundary layer, while other models find 'older air'. With a flux boundary condition, these differences would also appear. Therefore, we do not really get the point made by the reviewer: *"since the emissions cannot be stopped and must get out of the boundary layer daily"*. The issue is that some models tend to mix the boundary layer slower than other models, and thus simulate different gradients of emitted species. Most clearly, this is observed for the SF6 and 222Rn comparisons, both having flux boundary conditions, but models have a different rate at which the emissions are distributed vertically. To summarise:

With fixed emissions, an identical amount of tracer is mixed differently over the domain, depending on the mixing characteristics of the model. In the AoA protocol, the amount of tracer in the model intentionally depends on mixing. After translation to time, these differences can then be interpreted as different atmospheric mixing time scales. The challenges that we face are: (1) indeed, there are no known tracers with a fixed concentration boundary. However, the protocol is designed to simulate tracers from which transport timescales can be derived, and this advantage is explored here. (2) depending on the forcing volume and location, large gradients in the artificial AoA tracers may occur. Coping with large gradients is a known challenge for advection schemes. We discuss this issue in the manuscript in Section 3.6: Mapping the tropopause. Here we compare the tropopause derived from tracer 'e90' to the tropopause derived from AoA tracer 'surface'. In general, reasonable agreement is found, but differences are larger for some models. One tentative result is that models with less gradient conserving advection schemes (e.g. ACTM, Lin and Rood) show larger differences than the more gradient conserving models (like TM5, Russel and Lerner slopes scheme). In the revised manuscript we will discuss the issues with the protocol more comprihensively.

- *Missing references*: Thanks for pointing these out. We will refer to them in the revised manuscript.

- *Are the monthly mean mixing ratios 3D? Are the hourly station data 1D (profiles) or just layer 1?*: Yes, the monthly mean output is 3D. Modellers were asked to sample the model at 247 stations at hourly time intervals. At 119 station locations, profiles of concentrations and meteorological data were also requested. We will add some more information in the revised manuscript.

- *P8 L14 ? do you mean "adapted" instead of adopted?*: we did intend to use adapted here. Will be corrected.

- *P11 L29 ? is "IH" defined earlier? might be better just spelling it out everywhere, after all you did not define VT for vertical transport?*: IH is used 29 times in the manuscript, while vertical transport is used only 8 times. IH is defined on page 2.

- *P12 L13 ? please give real numbers - -how many years older?*: This is quantified in figure 9. We will add the range $0.10 - 0.17$ years.

- *S to N transport patterns*: This is an interesting issue. We speculate on page 22 the the asymmetry between Figures 3 and 4 is caused by a 'seasonal rectifier' effect, meaning that over NH landmasses the forcing is strongly coupled to the season. Due to wintertime stabilisation and strong mixing in the summer, the AoA metric shows a larger seasonal variation in the NH compared to the ocean-dominated SH. Indeed, we found much larger seasonal cycles in the time series that make up Figure 3,

compared to the time-series on which Figure 4 is based. How this relates to transport patters that transport the SH air masses to the NH (i.e. the coupling to monsoon circulations), remains to be explored and data to do so are available. We would like to restrict the current manuscript to a description of the first results.

- *The "oldest" tropospheric air noted here was also part of the Prather e90 paper, at least compare*: We will include a comparison in the revised manuscript.

- *Easter Island*: Typo will be corrected.

- *BL differences*: We are pretty sure that BL parameterisation differences are the cause. Instead of adding further analyses, we propose to remove "likely".

- *P15 L7 ? In addition to Diallo et al 2012, check the Prather 2008 PNAS paper that demonstrated the convergence of two schemes with increased vertical resolution.*: Thanks for pointing out this reference. We will include it in the revised manuscript.

- *P16 L7 ? Remember that the SF6 is flux-driven and may show different features than the AoA tracers.*: We will highlight this issue more strongly in the revised manuscript, also in response to the first point raised.

- *P18 L1 ? SF6 accumulation in the BL*: We have been careful here with the wording. We mention $SF_6$ accumulation over the source regions over land when refer to the top panel of Figure 8, in which a clear enhancement over the source regions is seen in the simulated SF6 fields.

-  P19 ? figures 9 and 10 : With Figure 9 we try to quantify the interesting asymmetry between the tracers 'NHsurface' and 'SHsurface'. Close to the equator the results become very sensitive to the way the model is sampled at the surface. This was illustrated with Figure 1. When forcing a tracer at the surface, e.g. in the SH, only the lowest 100 m of the grid-cel is forced (Table 1). After forcing, advective processes may transport AoA tracers into the box before sampling is performed. Close to the equator, large gradients in the mixing ratio of these NHsurface and SHsurface AoA tracers are expected, leading to results that depend on the sampling strategy of the models. Concerning the TOMCAT results: Indeed TOMCAT simulates the correct latitudinal gradient over the clean pacific (Figure 8, middle panel). However, the point we try to make is that this is a combined effect of slow vertical mixing (Figure 8, upper panel) and slow IH transport (quantified through the composite AoA in Figure 10). In contrast, NIES is also not "ball-park" in Figure 10. This is attributed to a combination of limited vertical mixing and fast IH transport. In that respect, the comparison to HIPPO data presented in the Appendix is interesting. Shifts to modeled mixing ratios needed to match the HIPPO data are largest for NIES and TOMCAT, but opposite in sign.

- *Please spell out IH and BL etc in the conclusions*: Will be done.

- *The Conclusions are a bit too speculative*: We agree on the fact that more work is needed to separate the effects of advection, convection, mass-fixers etc. We will modify the conclusions accordingly. Concerning the rectifier effect, we will try to explain this better in the revised manuscript. The bottom line is that when mixing is fast in the NH, the surface is in relatively fast contact with the free troposphere. Since inter-hemispheric transport proceeds most efficiently in the free troposphere (as seen in Figures 3 and 4), this implies transport of 'young' air to the SH. In winter, the situation reverses. Averaged over 11 years this leads to an average AoA that is younger than a situation with small seasonal variation in vertical mixing (i.e. the SHsurface tracer). To illustrate this, we build a simple three box model, with the lowest box representing the 'forced' surface layer. This box is in contact with box two, representing the boundary layer. Box two, in turn, is connected to box three, representing the free atmosphere. For this simple example, boxes are taken of equal size. The mixing time between the surface and boundary layer box is taken as 1 day. The mixing time between the boundary layer box and the free atmospheric box is taken as one week. To represent the NH, these times are modulated with a seasonal variation of 50%. To represent the SH, a smaller modulation of 5% is chosen. We now conducted the AoA experiments and calculated (in arbitrary units) the AoA in the NH and SH simulations in the three boxes (see Figure 1, upper panel). Logically, seasonal variations are larger in the NH case. If we now compare the running averages in the lower panel, we notice a small rectifier effect: The mean of the upper atmosphere SH boxes is systematically older than the corresponding NH boxes. This rectifier effect is caused by a covariation of the mixing and forcing. For the revised manuscript, we plan to include a worked out example with more realistic values as an Appendix.

[Figure]

Figure 1: Results of a simple three-box AoA experiment in which the mixing between the boxes is modulated with a seasonal cycle. See main text for a description of the experiment.

---

## Referee Report (RR1)

The manuscript have been improved but there are still some issues that need to be addressed. In the following here are my major points and general concerns:

**Major points:**

1. *We are totally agree in the method to proceed to assess the induced differences by different setup. I do also think that analysing the exact causes for the inter-model differences would lengthen the paper considerably. Therefore, I recommend to limit the analysis to these three model LMDz, TM5 and EMAC with their 7 simulations and the value more the utility of the method and its possible application to assess model transport, resolution and convection parameterisation biases. With LMDz simulations, the convection parametrisation issues can be assess as the difference between both runs is the convection scheme. With the three simulation of TM5, the vertical and horizontal resolution bias can be analysed. With EMAC runs, the resolutions issues can be analysed. The ACTM, NIES and TOMCAT differences can not be assessed with these runs and neither understood. Any explanation proposed here about this differences for three models is speculation. I strongly believe the bias in the reanalysis specially JRA-55 might play an important role in addition to the convection parameterisation proposed here. Without an explanation of a further analysis, these model results make the paper a bit incomplete. A second paper can written where the issues concerning these models (ACTM, NIES and TOMCAT) are carefully addressed in term of reanalysis, convection parameterisations, vertical and horizontal resolution issues.*

2. *The paper still need to be improved in presentation quality. There are several minor issues but important for quality of the paper. For instance, the use of term before defined them, inappropriate use of English lan-*

*guage, wrong use of citations (Wofsy, 2011), not homogenized definition (re-analysis, reanalysis, inter-comparison, intercomparion etc...) that should be improved. Some sentences are not formulated in good English. Maybe asking a help to native English speaker would be helpful to overcome these issues.*

**Minor points:**

1. *Abstract, line 4: ". . . to and in the stratosphere . . ." Please replace this by "into the stratosphere"*

2. *Abstract, line 8: Remove ". . . as used here, . . ."*

3. *Abstract, line 9: Remove the second ". . . over . . ."*

4. *Abstract, line 10: rephrase " We also find . . ."*

5. *Abstract, line 13-14: Please be consistent with word that are used in the text ". . .northern hemisphere. . ." or Northern Hemisphere. Same for Southern Hemisphere as well. Please check the same issues in the whole manuscript.*

6. *Page 2, line 5: ". . .mixing ratio distribution of CH4..." Replace by "... distribution of CH4 mixing ratio..."*

7. *Page 2, line 9: Boucher et al., 2009 can be cited here.*

8. *Page 2, lines 13-14: Remove "It emanates" and rephrase it to one sentence.*

9. *Page 2, lines 26-27: Please rephrase this sentence "The study concluded that models that show faster IH exchange for SF6, also exhibit smaller IH gradients in CH4."*

10. *Page 2, lines 33: Remove "....between..."*

11. ***Page 3, lines 23: Remove ''....just...''***

12. ***Page 2, lines 28: There is no need to define several time ''AoA''. It has been defined early, just use it all over the text. In addition, the age spectrum is well describe but there is not an explanation about the relation between AoA and age spectrum. In the text, you only use AoA. Therefore, one or two sentences to link the age spectrum to AoA is needed in this paragraph (27-30).***

13. ***Page 4, lines 6-10: There is no description of Table 1 about the different initialization of the tracers neither a clear description of the acronyms (NHSurface, etc... ) that are all over the manuscript. Please define these mask terminology and describe the setup here once.***

14. ***Page 7, line 6: It's important to be consistent. It's not ''... six CTMs...'' but ''3CTMs and 3 GCMs''.***

15. ***Page 7, lines 8: Replace ''... data of a reanalysis (European Centre for Medium-Range Weather Forecasts (ECMWF) ERA-Interim, or JMA).'' by ''. . . a reanalysis dataset, such as ERA-Interim (Dee et al.,2011) provided by European Centre for Medium-Range weather Forecasts and JRA-25 from Japanese Meteorological Agency (Onogri et al., 2007, Kobayashi et al., 2016)''. These defined terminology can be used later in the manuscript without redefined over again.***

16. ***Page 9, line 18: Replace ''... while ...'' by ''... when ...''***

17. ***Page 10, line 7 and line 19: Replace ''... Japanese Meteorological Agency ...'' by '' JRA-25''***

18. ***Page 9, line 32: Replace ''... ECMWF meteorological (re)analyses ...'' by ''... ERA-Interim reanalysis ...''***

19. ***Page 11, lines 16-19: Rephrase this sentence "Subsequently, the difference with respect to the South Pole is calculated as an 11-year time series with monthly time resolution. This is done to account for the model-data offset."***

20. ***Page 11, lines 26-28: Rephrase this paragraph. Remove " (and pressure field). change this sentence " Monthly mean mixing ratios (and pressure fields) of the participating models have been averaged zonally and over time, and converted to AoA." to "Converted into AoA, the monthly mean mixing ratios of the participating models have been averaged zonally and over time". Remove " contour plots of these ...." and replace it by "The lat-press cross-section of this AoA is ...". Should also use replace " AoA tracers NHSurface" by "AoA tracer from NHSurface" etc...***

21. ***Page 12, line 30: This is wrong "... signalling stratosphere-troposphere exchange (Holton et al., 1995)". This indicate "the upward transport of tropical tropospheric air into the stratosphere". In addition what is called "Deep convective mixing" is just "the tropical ascent" or "Deep convective transport".***

22. ***Page 13, line 1: Replace "... vertical mixing ..." by "... vertical transport ..."***

23. ***Page 13, line 3: Add "... from ..." after "tracers" and before "NHSurface"***

24. ***Page 13, line 20: "... north-east ..."***

25. ***Page 14, line 4: "... areas deep inland ..." by "deep inland areas"***

26. ***Page 14, line 8: Not define "... BL ..." but later in page 20. To what "This" belong? Rephrase it.***

27. ***Page 14, line 13: Rephrase this sentence "In Figure 5 the stratospheric AoA is plotted for all models from 100 hPa to the top of the atmosphere, averaged***

*over the period 20002010 (11 years). As expected, oldest AoA is found at the high-altitude poles". Start "Figure 5 shows the stratos...."*

28. ***Page 15, line 6: "... Wosfy, 2011" Wrong citation***

29. ***Page 15, line 9: "The lower panel..." of which figure?***

30. ***Page 19, line 12: remove " ...high level..."***

31. ***Page 20, line 11: To what belong "...,this...". Rephrase.***

32. ***Page 21, line 8-9: This is wrong "...and may become more apparent on the smaller spatial and temporal scales that are not part of this first analysis". This reanalysis bias are intrinsic to the scales.***

33. ***Figure 2 and 5: Please remove the pressure levels overplots.***

**References**

1. Eluszkiewicz, J., R.S. Hemler, J.D. Mahlman, L. Bruhwiler, and L.L. Takacs, 2000: Sensitivity of Age-of-Air Calculations to the Choice of Advection Scheme. J. Atmos. Sci., 57, 31853201, https://doi.org/10.1175/1520-0469(2000)057¡3185:SOAOAC¿2.0.CO;2

2. Boucher, O., Friedlingstein, P., Collins, B., and Keith P Shine, K. P.: The indirect global warming potential and global temperature change potential due to methane oxidation, Environ. Res. Lett., 4, 044007, doi:10.1088/1748-9326/4/4/044007, 2009.

3. Kobayashi, C., and T. Iwasaki (2016), Brewer-Dobson circulation diagnosed from JRA-55, J. Geophys. Res. Atmos., 121, 14931510, doi:10.1002/2015JD023476.

---

## Author Response (AR2)

**Detailed reply to reviewers comments (2)**

Maarten Krol et al.

July 16, 2018

**1 Major points**

First of all, we would like to thank the reviewer for the comments on the paper. The reviewer suggests to remove three models from the inter-comparison and still requests a detailed analysis of the model diversity (convection, horizontal and vertical resolution) based on LMDZ, TM5, and EMAC. Furthermore, the reviewer believes the bias in the reanalysis, especially JRA-55/25, might play an important role.

Concerning the first suggestion we strongly disagree with the reviewer. First, the presented large model diversity is the reality in a range of models that are applied for applications like inverse modelling and atmospheric chemistry simulations. This is a strong warning for the community: presented results may strongly depend on the model that is used for the analysis. Limiting the analysis to only three models and focussing only on an in-depth analysis of the underlying causes for diversity would considerably weaken this important message and drastically change the focus of the paper.

In our earlier reply we argued that our paper fits perfectly as a "Model Experiment Description Paper" in GMD: *Model experiment description papers contain descriptions of standard experiments for a particular type of model, such as might be used in a MIP. ..... In the case of papers describing MIPs, they should explain any specific project protocols, should highlight differences in the application of the protocol by the different groups, and should include sufficient descriptions/figures of model results to give an overview of the project. ..... The data sets must also be made available, and any code used to create the syntheses should also be made available.*

This does not imply that we disagree with the reviewer on the importance of further analysis. Our paper ends with "Most of all, further analysis should focus on the causes of the still large spread in the participating models". However, we do not agree with the principle that a reviewer should dictate how we should perform the analysis. In our view the manuscript presents interesting results and fits the GMD criteria. Removing three models at this stage is not an option for us.

That brings us to the second point brought up by the reviewer that the convective redistribution is not the most important driver for model diversity, but biases in the reanalysis data used for transport. Although we did not perform an in-depth analysis of re-analysis data, our discussion paper already triggered studies that indeed point to an important role for vertical transport (Patra et al., Improved Chemical Tracer Simulation by MIROC4.0-based Atmospheric Chemistry-Transport Model (MIROC4-ACTM)). We also down-played our claim considerably in the revised manuscript, and we agree on including a few lines one the possible role of reanalysis data (see minor comments below).

A final point made by the reviewer is about the use of the English language. We apologise for the annoying mistakes that have been spotted by the reviewer and we will correct them. The manuscript was scrutinised by native-speaking coauthors and we felt the language was of sufficient quality. However, taking on board the comment, we will carefully check the manuscript for the next and final versions.

**2  Minor points**

- *Abstract, line 4: ". . . to and in the stratosphere . . ." Please replace this by "into the stratosphere"* We specifically mean to say: to the stratosphere....and in the stratosphere. So we kept the sentence as it is.

- *Abstract, line 8: Remove ". . . as used here, . . .".* This was added to indicate that different model settings are possible. We prefer to keep this addition in.

- *Abstract, line 9: Remove the second ". . . over . . .".* Done

- *Abstract, line 10: rephrase "We also find . . .".* We wrote "We find" and kept it like this.

- *Abstract, line 13-14: Please be consistent with word that are used in the text "northern hemisphere" or Northern Hemisphere. Same for Southern Hemisphere as well. Please check the same issues in the whole manuscript.* Done.

- *Page 2, line 5: "mixing ratio distribution of CH4" Replace by " distribution of CH4 mixing ratio".* Done.

- *Page 2, line 9: Boucher et al., 2009 can be cited here.* Done.

- *Page 2, lines 13-14: Remove "It emanates" and rephrase it to one sentence.* Rephrased as "emanates naturally".

- *Page 2, lines 26-27: Please rephrase this sentence "The study concluded that models that show faster IH exchange for SF6, also exhibit smaller IH gradients in CH4.".* We rephrased as: "The study concluded that models with faster IH exchange for SF6 have smaller IH gradients in CH4".

- *Page 2, lines 33: Remove "..between..".* Done.

- *Page 3, lines 23: Remove "..just..".* Done.

- *Page 3, lines 28: There is no need to define several time "AoA". It has been defined early, just use it all over the text. In addition, the age spectrum is well describe but there is not an explanation about the relation between AoA and age spectrum. In the text, you only use AoA. Therefore, one or two sentences to link the age spectrum to AoA is needed in this paragraph (27-30)..* We added: "In the terminology of Holzer et al. (2000), AoA is defined as the mean transit time, with the age spectrum being the transit-time probability density function."

- *Page 4, lines 6-10: There is no description of Table 1 about the different initialization of the tracers neither a clear description of the acronyms (NHSurface, etc.. ) that are all over the manuscript. Please define these mask terminology and describe the setup here once..* We clearly wrote in discussing Table 1: "The mixing ratios of these tracers in the models are initialised to zero on January 1, 1988 and simulations are run to the end of 2014.". Also, Table 1 is used to define the forcing volumes. We do not see how we can be more clear.

- *Page 7, line 6: It's important to be consistent. It's not "... six CTMs..." but "3 CTMs and 3 GCMs"..* The reviewer is right. We modified this.

- *Page 7, lines 8: Replace "... data of a reanalysis (European Centre for Medium-Range Weather Forecasts (ECMWF) ERA-Interim, or JMA)." by ". . . a reanalysis dataset, such as ERA-Interim (Dee et al.,2011) provided by European Centre for Medium-Range weather Forecasts and JRA-25 from Japanese Meteorological Agency (Onogri et al., 2007, Kobayashi et al., 2016)". These defined terminology can be used later in the manuscript without redefined over again..* Good suggestion. Done.

- *Page 9, line 18: Replace "... while ..." by "... when ...".* Done.

- *Page 10, line 7 and line 19: Replace "... Japanese Meteorological Agency ..." by "JRA-25".* Done.

- *Page 10, line 32: Replace "... ECMWF meteorological (re)analyses ..." by "... ERA-Interim reanalysis ...".* Done.

- *Page 11, lines 16-19: Rephrase this sentence "Subsequently, the difference with respect to the South Pole is calculated as an 11-year time series with monthly time resolution. This is done to account for the model-data offset.".* We replaced with "To account for model-data offsets, 11 year time series with monthly resolution were constructed of the stations' mixing ratios with respect to the South Pole."

- *Page 11, lines 26-28: Rephrase this paragraph. Remove "(and pressure field). change this sentence " Monthly mean mixing ratios (and pressure fields) of the participating*

*models have been averaged zonally and over time, and converted to AoA." to "Converted into AoA, the monthly mean mixing ratios of the participating models have been averaged zonally and over time"..* We want to express that we first averaged the mixing ratio fields before we converted them to AoA. What the reviewer suggests would be incorrect. Apparently our phrasing was not clear. We rephrased as: "We created these averages by (i) averaging the monthly mean mixing ratios of the participating models zonally and over time (ii) converting the mean mixing ratios to AoA.

- *Remove "contour plots of these ...." and replace it by "The lat-press cross-section of this AoA is ..."..* We continue now as: "These latitude–pressure cross-sections are presented in".

- *Should also use replace "AoA tracers NHSurface" by "AoA tracer from NHSurface" etc..* We kept this as it was, since we define the meaning of NHSurface, etc. in Table 1.

- *Page 12, line 30: This is wrong "... signalling stratosphere-troposphere exchange (Holton et al., 1995)". This indicate "the upward transport of tropical tropospheric air into the stratosphere". In addition what is called "Deep convective mixing" is just "the tropical ascent" or "Deep convective transport".* We want to signal the tongues of older air that are an indication of older air from the stratosphere that enters the troposphere. We rephrased as: "bounded by tongues of older air, signalling intrusions of stratospheric air".

- *Page 13, line 1: Replace "... vertical mixing ..." by "... vertical transport ...".* Done.

- *Page 13, line 3: Add "... from ..." after "tracers" and before "NHSurface".* Again, we clearly defined the meaning NHSurface and other AoA tracers in Table 1. It therefore is a name, and "from NHSurface" would be incorrect.

- *Page 13, line 20: "... north-east ...".* Unclear what is meant here.

- *Page 14, line 4: "... areas deep inland ..." by "deep inland areas".* Done.

- *Page 14, line 8: Not define "... BL ..." but later in page 20. To what "This" belong? Rephrase it..* We rephrased as: "..that this type of local boundary layer (BL) mixing scheme ...

- *Page 14, line 13: Rephrase this sentence "In Figure 5 the stratospheric AoA is plotted for all models from 100 hPa to the top of the atmosphere, averaged over the period 2000-2010 (11 years). As expected, oldest AoA is found at the high-altitude poles". Start "Figure 5 shows the stratos....".* Rephrased.

- *Page 15, line 6: "... Wosfy, 2011" Wrong citation..* This reference is: "Wofsy (and not Wosfy) and the HIPPO science team and cooperating modellers and satellite teams (2011)" and we do not know how the journal wants to incorporate this reference.

- *Page 15, line 9: "The lower panel..." of which figure.* Now it reads: Panel (d) of Figure 6.

- *Page 19, line 12: remove "...high level...".* We removed "a high level of".

- *Page 20, line 11: To what belong "...,this...". Rephrase..* We guess the reviewer refers to line 9. We added "asymmetry" after this to make things clearer.

- *Page 21, line 8-9: This is wrong "...and may become more apparent on the smaller spatial and temporal scales that are not part of this first analysis". This reanalysis bias are intrinsic to the scales.* Here the reviewer strongly believes that inter-model differences (biases) are mainly caused by reanalysis data and not by differences in the parameterisations of convective transport. Although we did not perform an in-depth analysis of re-analysis data, our discussion paper already triggered studies that indeed point to an important role for vertical transport (Patra et al., Improved Chemical Tracer Simulation by MIROC4.0-based Atmospheric Chemistry-Transport Model (MIROC4-ACTM)). To trigger awareness of the role of the meteorological driver data, we added close to the end of the manuscript: "AoA studies with a single transport model and different meteorological driver data (e.g. ERA-Interim versus JRA-25) would reveal the potential role of the meteorological driver data on AoA biases."

- *Figure 2 and 5: Please remove the pressure levels overplots.* We do not agree with this suggestion, because the pressure level overplots provide the reader with extra information.

[revised manuscript text omitted]